# Global, quantitative and dynamic mapping of protein subcellular localization

**Daniel N Itzhak, Stefka Tyanova, Jürgen Cox, Georg HH Borner\***

Department of Proteomics and Signal Transduction, Max Planck Institute of Biochemistry, Martinsried, Germany

**Abstract** Subcellular localization critically influences protein function, and cells control protein localization to regulate biological processes. We have developed and applied Dynamic Organellar Maps, a proteomic method that allows global mapping of protein translocation events. We initially used maps statically to generate a database with localization and absolute copy number information for over 8700 proteins from HeLa cells, approaching comprehensive coverage. All major organelles were resolved, with exceptional prediction accuracy (estimated at >92%). Combining spatial and abundance information yielded an unprecedented quantitative view of HeLa cell anatomy and organellar composition, at the protein level. We subsequently demonstrated the dynamic capabilities of the approach by capturing translocation events following EGF stimulation, which we integrated into a quantitative model. Dynamic Organellar Maps enable the proteome-wide analysis of physiological protein movements, without requiring any reagents specific to the investigated process, and will thus be widely applicable in cell biology.

**\*For correspondence:** borner@biochem.mpg.de

**Competing interests:** The authors declare that no competing interests exist.

## Introduction

The hallmark of eukaryotic cells is their compartmentalization into distinct membrane-bound organelles. Protein function is critically determined by subcellular localization, as organelles offer different chemical environments and interaction partners. In order to regulate protein activity, many biological processes involve changes in protein subcellular localization. Prominent examples include the endocytic uptake of activated plasma membrane signalling receptors, to terminate the signalling process (*Jones and Rappoport, 2014*), and the nucleo-cytoplasmic shuttling of many transcription factors, to regulate their access to DNA (*Plotnikov et al., 2011*).

The ability to monitor changes in organellar composition would provide a powerful tool to investigate cell biological processes at the systems level. While transcriptomic (*Curtis et al., 2012*) and proteomic abundance profiling approaches (*Deeb et al., 2015*) have yielded valuable insights into changes in gene or protein expression, they lack the important spatial dimension. Microscopy-based approaches can provide spatial information on individual proteins (*Uhlen et al., 2015*), but are limited by the availability of specific antibodies, and are very labour-intensive for analysing complete proteomes (*Marx, 2015*). Genome-wide GFP-tagging in yeast circumvents the need for antibodies (*Huh et al., 2003*), but tags may inadvertently alter protein subcellular localisation, which is difficult to control for; in addition, serial imaging of cells for comparative purposes remains experimentally challenging (*Breker et al., 2013*).

Mass spectrometry-based proteomics has much enhanced our understanding of cellular composition (*Larance and Lamond, 2015*). Although sophisticated approaches for organellar proteomics have been available for over a decade (*Andersen et al., 2003*; *Christoforou et al., 2016*; *Dunkley et al., 2004*; *Foster et al., 2006*; *Gilchrist et al., 2006*; *Smirle et al., 2013*), there is

**eLife digest** The interior of every cell is highly organised, and contains many compartments, called organelles, that are dedicated to specific roles. Proteins are the tools and machines of the cell, and each organelle has its own set of proteins that it requires to work correctly. Each cell contains ten or more organelles, and several thousand different types of proteins. The exact location of proteins in the cell is important; once we know what compartment a protein is in, it is easier to narrow down what it might be doing.

The location of many proteins in a cell is unclear or simply not known. Moreover, since changing the location of a protein can change its activity, it is also important to be able to detect changes in the location of proteins under different circumstances, such as before and after drug treatment.

Itzhak et al. set out to develop a method that reveals the locations of all the proteins in a cell at any given time. The resulting technique maps the location of most of the proteins in a human cancer cell line and, in addition, determines how many copies of each protein there are. Combining these two types of information produces a model of the cell's architecture. Importantly, Itzhak et al. were able to compare such a model of the cell under normal circumstances to a model made after the cell had been stimulated with a growth factor. This revealed which proteins had changed location, identifying these proteins as important for the cell's response to the growth factor.

The new mapping method could be used in the future to analyse the anatomy of different cell types, such as nerve cells and cells of the immune system. Itzhak et al. also want to investigate the differences between healthy cells and cells from people with neurological disorders to understand how such diseases arise.

currently no proteomic method that allows global dynamic mapping of protein subcellular localization. The main reason for this deficiency is the high variability between spatial proteomics experiments, which renders the identification of genuine organellar transitions very difficult (*Gatto et al., 2014*).

Here, we have developed a rapid proteomic profiling workflow for the generation of highly reproducible organellar maps. We use the method to assemble a comprehensive database of protein subcellular localization and abundance information from HeLa cells, allowing us to build a quantitative model of cellular anatomy. We then apply organellar maps to capture the protein translocation events triggered by EGF stimulation, demonstrating the dynamic capabilities of our approach.

## Results

### Organellar maps through fractionation profiling

The principle of our approach is to separate organelles partially with a minimum number of fractionation steps, and to generate organellar profiles by high-accuracy quantification of each fraction against an invariant reference. Metabolically labelled HeLa cells, SILAC light or heavy (*Ong et al., 2002*), were mechanically lysed following gentle hypo-osmotic swelling (*Figure 1A*). Damage to organelles was minimal, as assessed by leakage of lumenal contents (*Figure 1—figure supplement 1A*). Post-nuclear supernatant from light cells was fractionated by a series of five differential centrifugation steps, whereas a total organellar 'reference' fraction was obtained in a single centrifugation step from heavy post-nuclear supernatant. This procedure is highly reproducible, as assessed by protein recovery (*Figure 1—figure supplement 1B,C*). Each light sub-fraction was then combined with an equal amount of the heavy reference, subjected to tryptic digest and analysed by LC-MS/MS. For each protein, we obtained an abundance distribution profile across the sub-fractions. In a typical experiment, approximately 4500 proteins were profiled. Proteins associated with the same organelle have similar profiles, and organelles can be distinguished from one another (*Figure 1B*). In parallel, the global distributions of proteins across the nuclear, organellar, and cytosolic fractions were obtained by label-free quantification mass-spectrometry, typically covering 8000 proteins (*Figure 1C*).

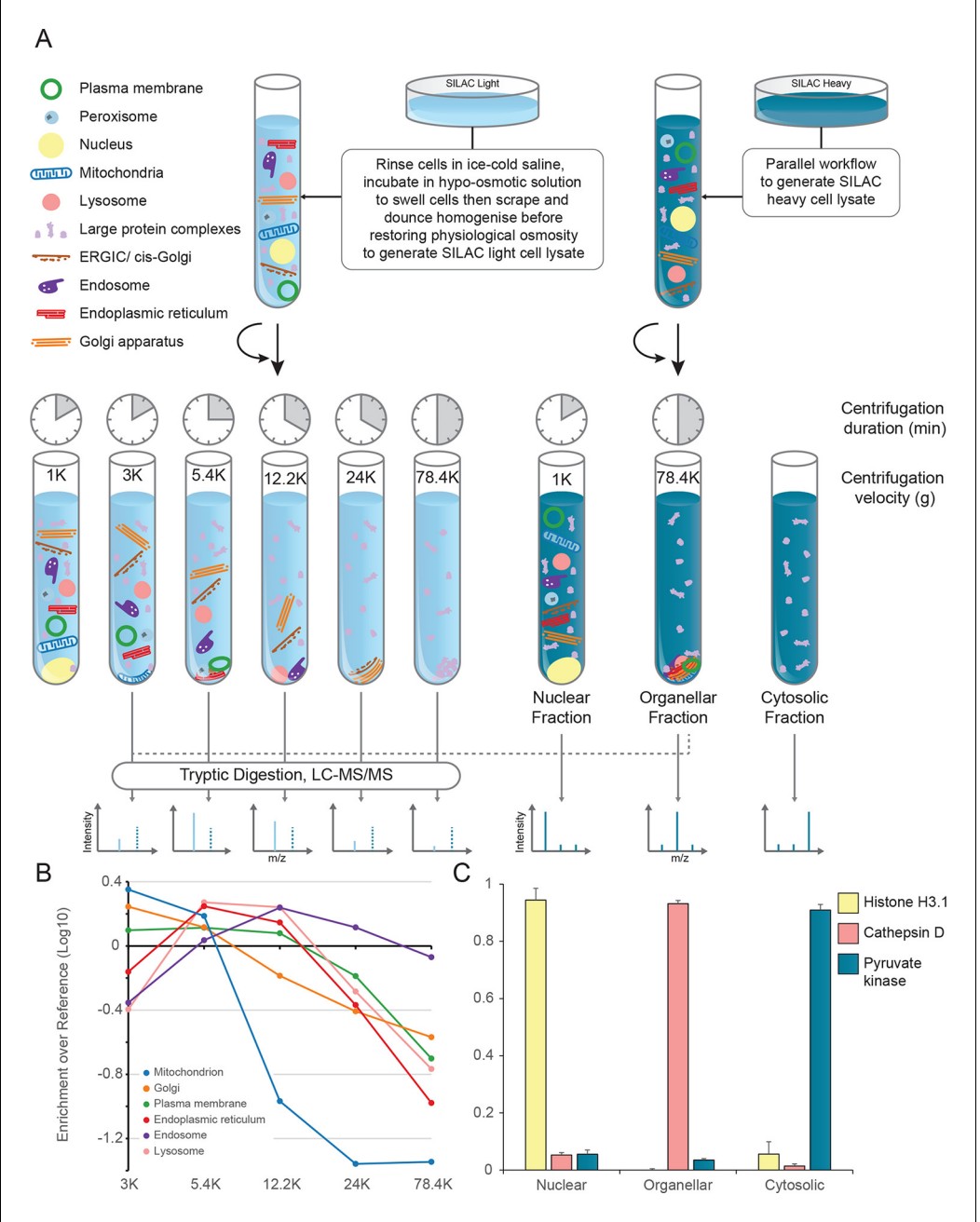

**Figure 1.** Generation of organellar maps through fractionation profiling. (**A**) Metabolically labelled HeLa cells were mechanically lysed to release organelles. Light labelled lysate was then subjected to differential centrifugation at the indicated speeds (RCF$_{MAX}$) and times (in minutes). Heavy-labelled lysate was centrifuged twice, once at low speed to generate a nuclear-enriched pellet, and again at high speed to generate the organellar pellet; the supernatant was the cytosolic fraction. The heavy organellar 'reference' fraction was combined with equal protein amounts of each of the five light membrane sub-fractions and analysed by mass spectrometry, generating SILAC ratios for each protein in all fractions. (**B**) The SILAC ratios were converted to enrichment over reference. Median values of organellar marker proteins were plotted, showing clearly distinct profiles. (**C**) In a parallel analysis, the heavy-labelled nuclear, organellar and cytosolic fractions were subjected to label-free mass spectrometric analysis, revealing the global distribution of proteins across these three fractions. Examples of normalized profiles of marker proteins for the nucleus (Histone H3), lysosome (Cathepsin D) and the cytosol (Pyruvate Kinase) are shown. Bars show mean ± SD (n = 6). Please refer to *Figure 1—figure supplement 1* for organellar leakage analysis and evaluation of fractionation yield reproducibility.

*Figure 1 continued on next page*

*Figure 1 continued*

The following figure supplement is available for figure 1:

**Figure supplement 1.** Organellar leakage analysis (A) and fractionation reproducibility (B, C).

Following this scheme, we prepared six replicate fractionations, in batches of two, on three different days. We first considered the set of 3766 proteins common to all replicates, and applied principal component analysis (PCA) to their abundance profiles. The PCA scores plot was then overlaid with established organellar markers (*Supplementary file 1*), which clustered into distinct regions of the plot (*Figure 2*). Resolved compartments included plasma membrane, endoplasmic reticulum, ERGIC, Golgi apparatus, endosome, lysosome, peroxisomes and mitochondria, as well as a cluster of diverse large protein complexes, such as ribosomes and proteasomes. Closer inspection of the marker proteins suggested further sub-organellar resolution, revealing a partial divide between ER

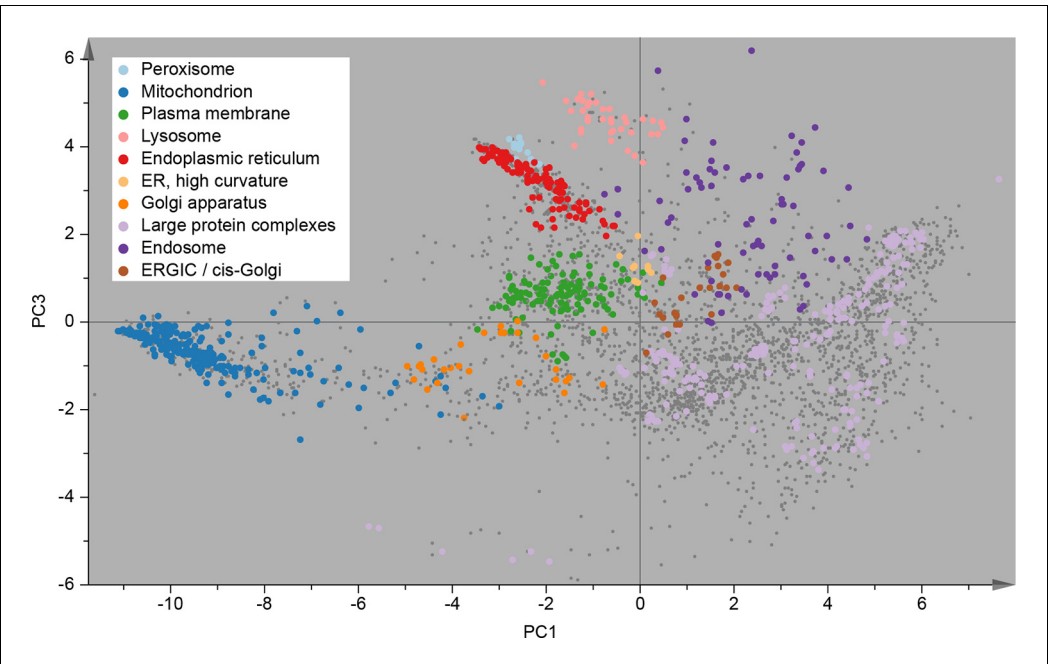

**Figure 2.** Visualization of an organellar map. Thirty SILAC ratios from six replicate fractionation experiments were combined and subjected to principal component analysis to achieve dimensionality reduction. Projections along the first (x-axis) and third (y-axis) principal components (PCs) provided the optimal separation of clusters. Each scatter point represents a protein. Proximity of proteins indicates similar fractionation behaviour. Marker proteins for organelles are coloured as indicated in the legend, and reveal clustering of proteins belonging to the same organelle. Non-marker proteins are shown as small grey dots. PCs 1–3 account for 64%, 21%, and 12% of the variability in the data, respectively. Please note that the actual resolution of the map is much higher than is apparent in this 2D representation of the full 30-dimensional data set, and most of the seemingly overlapping clusters are in fact separated. Please refer to *Figure 2—figure supplement 1* for more detailed cluster annotation, and overlays with external protein sequence feature predictions. *Figure 2—figure supplement 2* shows the reproducibility analysis of six replicate organellar maps. The complete organellar assignments, spatial and abundance information can be found in *Supplementary file 1* (compact format) and *4* (interactive database).

The following figure supplements are available for figure 2:

**Figure supplement 1.** Organellar map with full cluster annotation (A); overlay of an organellar map with external protein sequence feature predictions (B).

**Figure supplement 2.** Reproducibility analysis of organellar maps.

membrane and lumen, as well as division of mitochondria into matrix, inner membrane, and outer membrane (*Figure 2—figure supplement 1A*). For independent validation, we overlaid the scores plot with UniProt annotation for subcellular targeting features, including signal peptides, mitochondrial transit peptides and transmembrane domains, and observed near-complete agreement with our maps (*Figure 2—figure supplement 1B*). To assess the reproducibility of the method, we next analysed the six individual maps by PCA; all had very similar patterns (*Figure 2—figure supplement 2A*), with organellar clusters occupying stable positions between maps. The SILAC ratios of replicate fractions were also highly reproducible (average correlation > 0.95; *Figure 2—figure supplement 2B*).

For the rigorous assignment of proteins to organellar clusters, we used a support vector machine (SVM)-based supervised learning approach. Briefly, SVMs allow non-linear separation of clusters (*Varmuza and Filzmoser, 2009*). Optimal boundaries between organellar clusters are determined using marker proteins, with cross-validation to prevent over-fitting. Non-marker proteins falling within the boundaries of a particular cluster are then assigned to that organelle. Since a suitable canonical set of organellar markers was not available, we manually curated a set of over 1000 proteins (*Supplementary file 1*). We chose markers based on their expression in HeLa cells and their well-documented (and ideally unimodal) localization to a particular organelle. Clustering of these markers was visually confirmed with several PCA-based pilot maps (not included in this study). Where necessary, we specifically chose further established markers near the edges of organellar clusters, as these are particularly important for defining boundaries. We applied SVM classification to all six maps individually. The mean prediction accuracy for marker proteins was 94.7% (with full cross validation), demonstrating the exceptionally high level of organellar resolution achieved (*Supplementary file 2*). While marker prediction accuracy does not provide a direct measure of overall prediction accuracy, it nevertheless serves as a useful estimate (see Methods for further details). The average proportion of identical organellar assignments between maps, referred to as concordance, was 93.7% for all proteins, and >98% for three-quarters of the proteins (*Figure 2—figure supplement 2C*).

Collectively, these data show that fractionation profiling is effective for generating high resolution organellar maps. The remarkable level of reproducibility enables comparative applications (see below).

## A database of protein subcellular localization

We combined the predictions from the six replicate maps into a single output (see Methods for details). In total, we derived organellar profiles for 5265 proteins, of which 2423 were assigned to 9 membranous organelles, with 96.5% of marker proteins predicted correctly (92.7% average per membrane-bound organelle; *Table 1*). To validate the novel predictions, we removed the organellar markers from the set, and annotated the remaining proteins with UniProt subcellular localization information. A Fisher's exact test showed that for eight of the nine compartments, the most significantly enriched localization term corresponded to our own organellar classification (*Supplementary file 3*). Furthermore, we compared our mitochondrial predictions with the Mito-Carta database of experimentally validated mitochondrial proteins (*Calvo et al., 2016*); the overall concordance was 97% (92.3% for non-marker proteins). These data provide strong independent support for the high quality of our organellar assignments.

Organellar maps deliberately exclude the cytosolic fraction, since numerous peripheral membrane proteins have a soluble as well as a membrane-bound pool. Inclusion of the cytosol in the maps would reveal which proteins are predominantly cytosolic but sacrifice information on the precise localization of the membrane-associated fraction. Therefore, the maps were augmented by an auxiliary workflow, which reveals the nuclear-organellar-cytosolic distribution (*Figure 1C*). In total, this global profile analysis extends to 8710 proteins, including 1999 cytosolic, 1133 nuclear, and 672 nucleo-cytosolic proteins (*Supplementary file 1*).

We combined all data into a database, which contains three layers of information. At the global level, it includes the distribution of each protein between nuclear, organellar, and cytosolic pools, as well as copy numbers per cell and cellular concentrations (calculated with the 'proteomic ruler' approach [*Wisniewski et al., 2014*]). At the organellar level, predictions of compartment associations are provided, with confidence scores. Furthermore, maps have high local resolution; this third level of information provides the 'neighbourhood' of a protein, revealing which other proteins have

**Table 1.** Prediction output and performance of HeLa organellar maps. The table shows the combined organellar prediction output from six replicate maps from HeLa cells. Prediction performance is judged by the proportion of correctly assigned organellar marker proteins. Please also refer to **Supplementary file 1** (compact format) and **4** (complete database), which contain detailed information for all 8710 proteins covered in this study, including nuclear and cytosolic predictions. **Supplementary file 2** shows the performance of each individual map.

| Compartment | Number of marker proteins | Correctly predicted markers | | All proteins predicted in this compartment |
| --- | --- | --- | --- | --- |
| | | Number | % | |
| Endosome | 85 | 75 | 88.2% | 304 |
| ER | 127 | 127 | 100.0% | 530 |
| ER, high curvature | 11 | 11 | 100.0% | 45 |
| ERGIC/cisGolgi | 26 | 25 | 96.2% | 73 |
| Golgi | 33 | 29 | 87.9% | 190 |
| Lysosome | 43 | 41 | 95.3% | 88 |
| Mitochondrion | 242 | 239 | 98.8% | 658 |
| Peroxisome | 21 | 15 | 71.4% | 25 |
| Plasma membrane | 127 | 123 | 96.9% | 510 |
| All organellar proteins | 715 | 685 | 95.8% | 2423 |
| Average per organelle | | | 92.7% | |
| Large Protein Complexes | 361 | 353 | 97.8% | 2739 |
| Total | 1076 | 1038 | 96.5% | 5162 |

similar fractionation profiles. In many cases, this allows identification of stable protein complexes. The database is accessible via a web interface (www.MapOfTheCell.org), and as an interactive Excel file (**Supplementary file 4**); **Supplementary file 1** contains a compact summary of the organellar predictions and copy numbers. The website allows visual exploration of the individual organellar maps.

## Quantitative anatomy of a HeLa cell

Combined knowledge of protein subcellular localization and abundance enables construction of a model of HeLa cell composition. We calculated the protein mass of each organelle by multiplying the molecular weights of constituent proteins by their estimated copy numbers (**Figure 3**). This revealed that the endomembrane system contributes approximately 16% to total cellular protein mass, dominated by mitochondria (6.6%), ER (4.4%), and plasma membrane (3.1%), with relatively minor contributions from endosomes, lysosomes, peroxisomes and Golgi (**Figure 3A**). The mitochondria, ER and plasma membrane are themselves dominated by a few highly abundant proteins (**Figure 3B**). In each case, the 20 most abundant proteins constitute at least 40% of organellar protein mass (**Figure 3C–E**, and **Supplementary file 5**). For example, the most abundant plasma membrane protein is the 4F2 cell-surface antigen heavy chain (SLC3A2), with 15 million copies/cell. This versatile protein can heterodimerize with several other proteins (eg SLC7A5, another very abundant protein, three million copies) to form amino acid transporters. This predominance probably reflects the adaptation of HeLa cells for fast nutrient uptake to support rapid growth. Supporting this view, all plasma membrane transporters combined (40 million copies) contribute approximately 25% of the total compartment protein mass. Other integral membrane proteins (such as adhesion and signalling receptors) contribute ~30 million copies. Assuming a cell surface area of ~1600 μm$^2$ typical of adherent HeLa cells (**Fisher and Cooper, 1967**) yields an estimated density of 4–5 integral membrane proteins per 100 nm$^2$, in excellent agreement with a previous biochemically derived estimate of 3 for baby hamster kidney (BHK) fibroblasts (**Quinn et al., 1984**). Within the endoplasmic reticulum, proteins involved in protein folding and quality control predominate (20% chaperones, 10% protein disulfide isomerases). A similar abundance of chaperones was observed in the mitochondria (14%), which exceeds the collective contribution of citric acid cycle components (9%). We detected

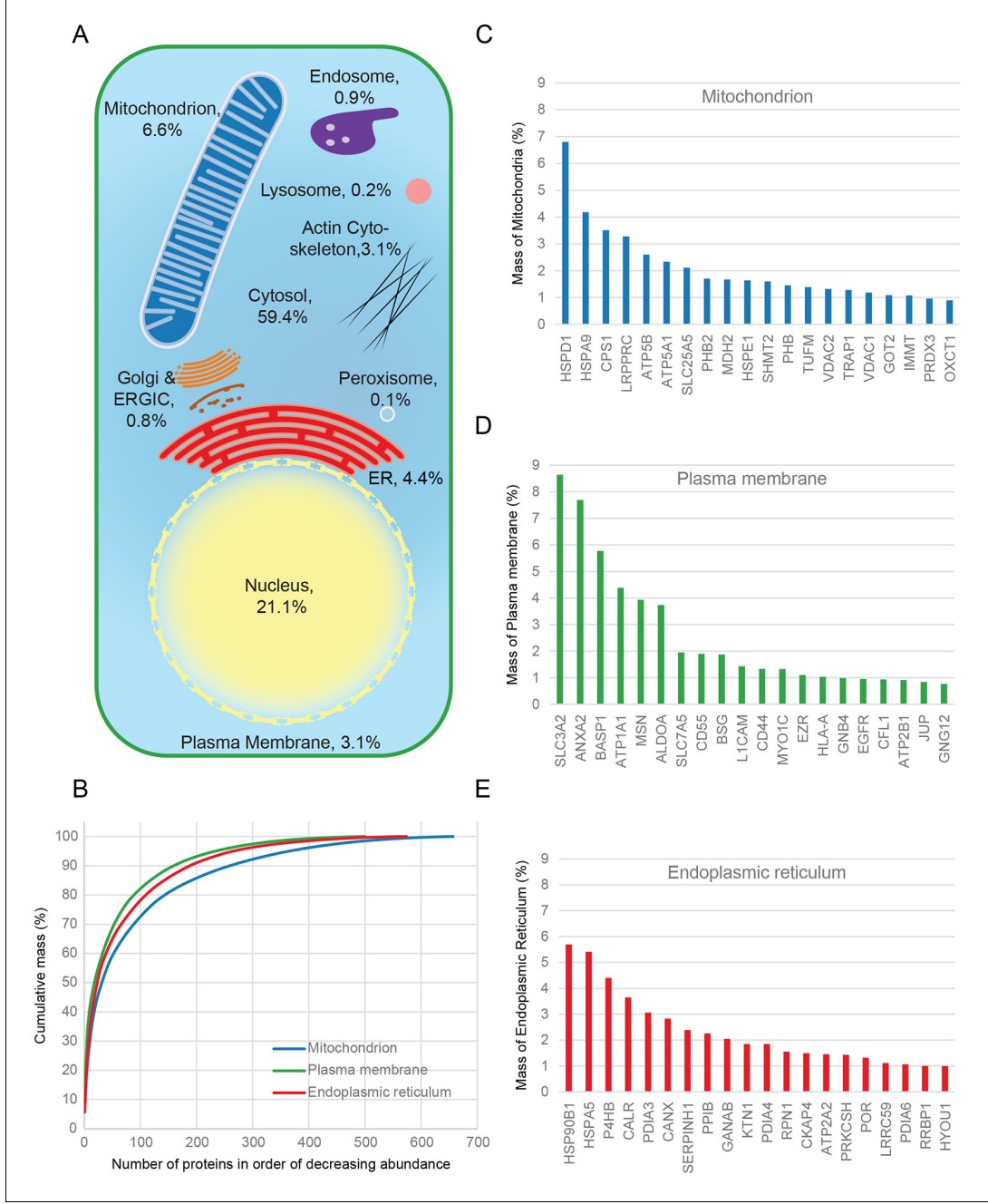

**Figure 3.** Quantitative anatomy of a HeLa cell. (**A**) Schematic diagram of a cell where compartments are approximately scaled by their relative contributions to total cell protein mass (*not by their volumes*). All membranous organelles combined (excluding the nucleus) contribute ca. 16%. For comparison, ribosomes and proteasomes contribute 6% and 1.3%, respectively. The proportion of the ER fraction would increase from 4.4% to ca. 5.4% if attached ribosomes were included. (**B**) Proteins of major organelles were ranked in order of decreasing abundance, and plotted against their cumulative mass. Very few proteins contribute the majority of organellar protein mass in all three cases. (**C–E**) Top 20 most abundant proteins in each of the three major organelles, plotted against their contribution to protein organelle mass. The complete quantitative composition of ER, mitochondria, and plasma membrane are shown in *Supplementary file 5*.

the five members of the mitochondrial ATP synthase $F_0$ catalytic complex with the expected stoichiometry of ~3:1 for subunits A/B to C/D/E, and estimate the number of complexes at ~3 million per cell (5% of mitochondrial protein mass). Thus, a picture of HeLa cell anatomy emerges from the quantitative subcellular localization information.

## Systems-wide detection of protein translocation events – dynamic organellar maps

The very high reproducibility of our approach opens the possibility to compare maps under different physiological conditions, to identify protein translocation events. To test this, we investigated the well-characterized process of epidermal growth factor receptor (EGFR) uptake. Following stimulation with EGF, EGFR autophosphorylates, binds downstream factors, and is rapidly endocytosed from the plasma membrane to an endosomal compartment (*Jones and Rappoport, 2014*). The translocation process is readily imaged using fluorescently-labelled EGF (*Figure 4A,B*). We prepared organellar maps from untreated (control) HeLa cells and from HeLa cells continuously stimulated with EGF for 20 min, in biological triplicate (*Figure 4C,D*; *Figure 4—figure supplement 1* provides a schematic of the experimental design; *Figure 4—figure supplement 2A* shows all six maps). Overall map morphology from treated and control cells was unchanged, however EGFR, which localized to the plasma membrane cluster in control cells, was localized to the endosomal cluster upon EGF treatment, as expected. To identify subcellular translocations in an unbiased manner, we developed a two-stage statistical analysis. For each protein the magnitude of translocation (Movement score) as well as the consistency of the direction of the translocation across biological repeats (Reproducibility score) were assessed. The two metrics were then combined in a 'MR' plot (*Figure 4E,F*) to identify proteins undergoing consistent translocations. To derive stringent score cut-offs, we took advantage of the maps used to generate our subcellular localization database (*Figure 2—figure supplement 2A*). We treated these six maps as a mock experiment in which we would not expect to detect any specific changes, by assigning three maps as 'controls' and three as 'mock-treated'. We determined the most stringent score cut-offs from the MR plot of the mock experiment by defining a region where no false positives were obtained. Applying these cut-offs to the EGF treatment experiment identified four proteins as significantly translocating; EGFR, GRB2, SHC1 and PKN2. Both GRB2 and SHC1 are recruited to EGFR upon EGF stimulation and constitute the first step in EGFR signaling (*Oda et al., 2005*). Inspection of the maps (*Figure 4C,D*, *Figure 4—figure supplement 2A*), and classification with support vector machines (*Supplementary file 6*) showed that all proteins had moved to the endosome/lysosomal compartment, as expected. Therefore, our approach correctly and specifically identified the major translocation events following EGF stimulation. For a deeper exploratory analysis, we then relaxed score cut-offs to allow an FDR of 10%, and identified 14 further significantly translocating proteins (*Supplementary file 7*). These included numerous known downstream targets of EGF, such as RPS6KA3, PIK3C2B and ROCK2, as well as several new candidates (see Discussion). These results validate the use of dynamic organellar maps for the systematic detection of subcellular translocation events.

In addition to intra-organellar translocation events, EGF signalling also involves cytosol/membrane as well as cytosol/nuclear transitions. To capture these events, we compared the abundance of proteins in membrane, nuclear and cytosolic fractions (as prepared in *Figure 1C*) from control and EGF-stimulated cells, based on label-free quantification (*Cox et al., 2014*) (*Figure 4—figure supplement 2B*). Stringent FDR controls were derived using a mock experiment of our six database maps, as above, identifying 26 significant changes in the EGF experiment (*Supplementary file 7*, and *Figure 4—figure supplement 3*). In agreement with the organellar maps, we detected substantial recruitment of GRB2 and SHC1 to membranes. In addition, we observed recruitment of CBL and UBASH3B, which are also known to bind to activated EGFR (*Grovdal et al., 2004*; *Raguz et al., 2007*). Consistent with that, CBL and UBASH3B were not detected in control maps but were found in individual EGF-treated maps, in the endosome/lysosome (*Figure 4—figure supplement 2A*). Among others, we identified the known translocation of RPS6KA3 into the nucleus, as well as a surprising number of transcriptional regulators leaving the nucleus (*Supplementary file 7*). These included ZCCHC8 and RBM7, the unique components of the nuclear exosome targeting (NEXT) complex (*Lubas et al., 2011*), which targets the exosome to promoter upstream transcripts (PROMPTS) for their degradation. Therefore our data suggest that EGF may induce modulation of the non-coding transcriptome.

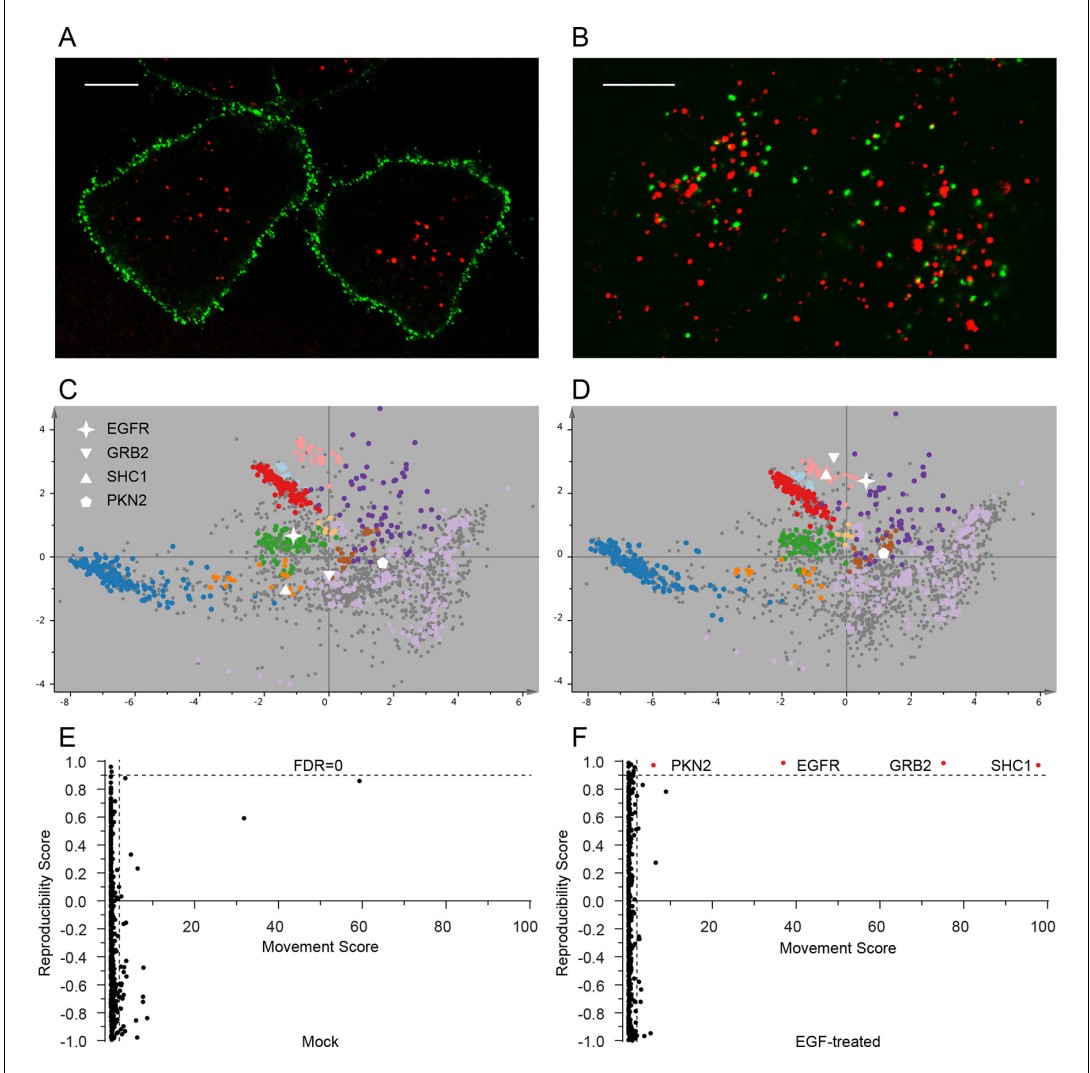

**Figure 4.** Dynamic organellar maps reveal protein localization changes following EGF stimulation. (**A, B**) Fluorescently tagged EGF (green) was pre-bound to HeLa cells on ice, and imaged by confocal microscopy. Lysosomal compartments were visualized with Lysotracker (red). Most of the EGF was at the cell surface (**A**). Cells were then shifted to 37°C, and incubated for 30 min. EGF had been cleared off the cell surface, and localized predominantly to an endosomal compartment, with little lysosomal co-localization (**B**). Scale bars = 10 μm. (**C**) Organellar maps were prepared from untreated HeLa cells, and (**D**) from cells following 20 min of continuous stimulation with 20 ng/ml EGF. The translocation of the EGFR receptor (star symbol) from plasma membrane to endosomes was captured. Colours indicate organelles as in *Figure 2*. Maps show the combined data from three replicates each. (**E, F**) Unbiased identification of significant translocation events triggered by EGF stimulation. Each protein is scored for magnitude of translocation (M score, x-axis) and reproducibility of translocation direction (R score, y-axis) across the three replicates. A MR plot reveals significant translocations in the top right quadrant. Score cut-offs for FDR-control were determined by analysis of a triplicate mock experiment where no genuine translocations are expected (**E**). Ultra-stringent cut-offs (corresponding to an FDR of 0) were then applied to the EGF treatment experiment (**F**). Four significant translocations were detected, including EGFR and two known binding partners, GRB2 and SHC1. As the maps in **C, D** reveal, all move to the endolysosomal cluster. *Figure 4—figure supplement 1* provides a schematic of the experimental design. Please refer to *Figure 4—figure supplements 2* and *3* for further in-depth analysis of protein localization changes following EGF stimulation.

The following figure supplements are available for figure 4:

**Figure supplement 1.** Dynamic organellar maps (EGF-treatment) – overview of the experimental workflow.

**Figure supplement 2.** Protein localization changes following EGF stimulation.

**Figure supplement 3.** Global protein distribution profile changes induced by EGF treatment.

## Quantitative modelling of EGF-triggered subcellular translocations

Finally, we combined the identified translocation events with our estimates of absolute protein abundances. For each translocating protein, we calculated the number of molecules in cytosol, nuclear, and organellar fractions, before and after EGF treatment. Differences were then interpreted as the number of proteins moving between compartments (summarized in *Supplementary file 7*). For example, our data show a significant overall loss of EGFR upon EGF treatment (from 700,000 to 620,000 copies per cell, p=0.0022), suggesting that a proportion of endocytosed EGFR has already been degraded in lysosomes. Approximately 500,000 copies of GRB2 are recruited onto endosomes/EGFR, suggesting a stoichiometry of ~1:1 with EGFR. In contrast, CBL (10,000 copies) and UBASH3B (30,000 copies) are recruited sub-stoichiometrically, as would be expected of enzymatically acting proteins. SHC1 (100,000 copies) is also recruited sub-stoichiometrically. The cell loses three quarters of its Vasorin (a negative regulator of TGFB signalling; 30,000 copies), most likely through plasma membrane shedding (*Malapeira et al., 2011*). Over 300,000 copies of the actin regulator Palladin are released into the cytosol, and the Rho-effector PKN2 is shifted to endosomes, indicating major cytoskeletal rearrangements. Our data thus begin to provide a quantitative, integrated view of EGF-triggered subcellular translocations at the protein level (*Figure 5*).

## Discussion

We have developed and applied a powerful new method for making quantitative organellar maps, to generate an extensive database of human protein subcellular localizations and organellar composition. Furthermore, the ease and reproducibility of our approach permits comparative applications and process modelling, as we demonstrate using EGF signalling.

### The HeLa spatial proteome

Here, we provide localization information for 8710 proteins in HeLa cells. The database is accessible via an Excel file (*Supplementary file 4*) and a website (www.MapOfTheCell.org), which provide complementary features for analysing the data. Both contain information on protein abundance (copy numbers per cell), global cellular distributions (eg cytosolic vs membrane pools), and predicted organellar associations. In addition, the website offers visualization and interactive exploration of the maps. *Supplementary file 4* provides an extra local 'neighbourhood analysis' identifying proteins with highly similar fractionation profiles (useful for identifying potential protein complexes), and also allows easy annotation of whole protein families via its batch submission option.

The complexity of the HeLa proteome has been estimated at around 10,000 proteins (*Beck et al., 2011*; *Nagaraj et al., 2011*); a substantial proportion of this is covered by our database. Importantly, it accounts for the vast majority of protein cell mass (as can for example be seen from the cumulative mass plots in *Figure 3B*, which all reach a stable plateau); further identifications would mostly correspond to low abundance proteins, with minimal contributions to organellar composition. In this respect, our database approaches comprehensive coverage, and offers a quantitative view of cellular architecture (*Figure 3*). The relative sizes of organelles differ significantly between cell types; the approach presented here allows a comparatively rapid characterization at a level previously only achievable through extensive morphological studies. A future comparison of different cell types will substantially enhance our understanding of cellular identity, by uncovering universal features and specific adaptations. In addition to the organellar level, it will also give new insights for individual proteins, by revealing cell- or species-specific localization differences, and thus potentially new regulatory or functional aspects.

### Accurate, quantitative and reproducible organellar maps

The profiling approach presented here maximizes speed and simplicity of the subcellular fractionation procedure. This ensures reproducibility, and at the same time keeps organelles as intact as possible. Since the preparative aspects are straightforward, several fractionations can be carried out in parallel on the same day, allowing multiplexing and complex experimental designs. Relative to the previous LOPIT approach (localization of organelle proteins by isotope tagging; *Christoforou et al., 2016*), our fractionation protocol is five times faster (4 hr vs ~20 hr), and requires an order of magnitude less starting material ($10^7$ cells vs $10^8$ cells). Most importantly, our

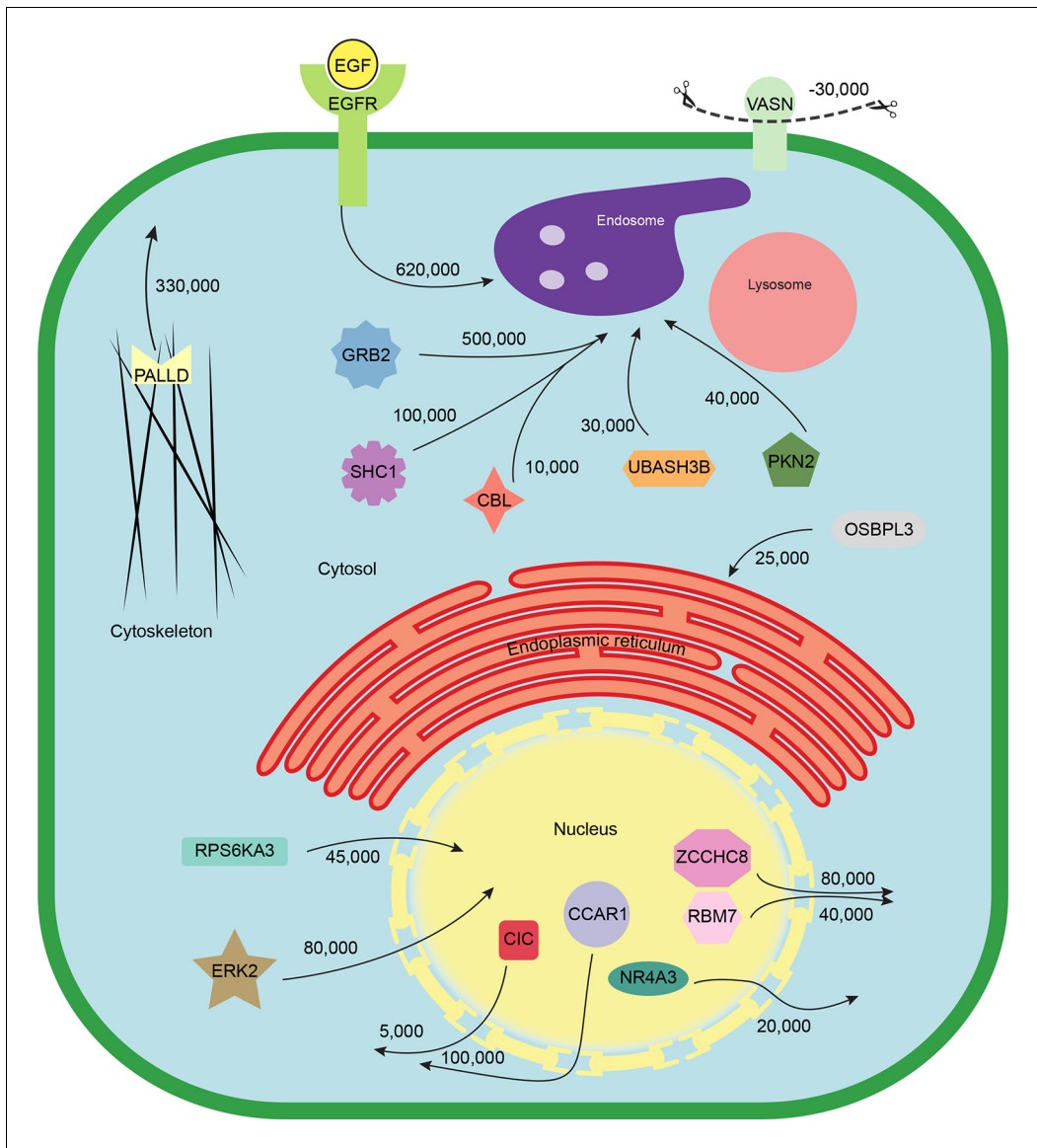

**Figure 5.** Quantitative mapping of EGF-triggered subcellular translocation events. Summary of key protein translocations in HeLa cells following 20 min of continuous stimulation with EGF. All depicted changes were detected by organellar maps in this study; they include numerous previously known as well as novel translocation events. Numbers on arrows indicate how many copies of a protein undergo the indicated movement (per cell). These estimates were also calculated from the mass spectrometry data, using the proteomic ruler approach (*Wisniewski et al., 2014*). *Figure 4—figure supplement 3* and *Supplementary file 6* (interactive database) and *7* (compact summary) show additional translocations not included here.

method can be used comparatively, and also offers quantitative data on protein abundance; a comparative application of LOPIT has yet to be demonstrated. The peptide labelling strategy allows very flexible use of LOPIT; our method requires metabolic labelling (SILAC), currently rendering it most suitable for dividing cells in culture. However, an application of fractionation profiling to mammalian tissues is possible, since mice can be kept on a SILAC diet (*Zanivan et al., 2012*); alternatively, a representative mix of SILAC-labelled cell lines may be used to generate the reference fraction (SuperSI-LAC approach [*Geiger et al., 2010*]). In addition, mass tagging is in principle compatible with our approach, too, and may thus extend its range of applications in future. A detailed comparison of the methods' relative advantages and requirements is presented in *Supplementary file 8*.

Our organellar assignments are in excellent agreement with independent external data (*Figure 2—figure supplement 1*, *Supplementary file 3*). Furthermore, we made a direct comparison with a recent analysis of the mouse stem cell spatial proteome using LOPIT (*Christoforou et al., 2016*). 2397 homologous proteins were classified in both studies, of which 2196 had identical compartment predictions (91.6%; *Supplementary file 3*). This exceptionally high level of agreement, across species and cell types, reciprocally supports the very high accuracy of predictions in both datasets.

Organellar maps based on subcellular fractionation profiles reflect protein steady state localizations. Proteins predominantly associated with a single organelle have closely matching profiles, and can be assigned unambiguously. In contrast, proteins equally split over two (or more) compartments have mixed profiles, which may be difficult to interpret (*Gatto et al., 2014*). Here, we assign each protein to the most likely compartment, but potential secondary assignments are also indicated (*Supplementary file 4*). Furthermore, our two-tiered profiling approach considerably alleviates the dual-localization problem, by separating organellar predictions from quantifying a protein's nuclear, cytosolic and membrane pools. This allows, for example, the accurate characterization of nuclear-cytosolic shuttling proteins: instead of showing an ambiguous 'in between' state, our approach precisely determines how these proteins are distributed over the two compartments. Similarly, for proteins with a cytosolic and an organellar pool, it allows quantification of the distribution, in addition to identification of the membrane compartment. Of note, our dynamic implementation of maps is generally unaffected by multiple localization difficulties, since we uncouple the detection of protein translocations from organellar assignments (*Figure 4*). Thus, our approach allows the identification of translocation events, even if they only involve partial organellar transitions.

## Dynamic organellar maps applied to EGF signalling

Here, we have used organellar maps to analyse cellular events following EGF stimulation. We correctly captured the endosomal transition of EGF receptor, and recruitment of signalling adaptors. Remarkably, the translocations were detected with extremely stringent FDR control, using cut-offs where we expect no false positives. This supports that our approach is capable of identifying translocation events *de novo*, without having to filter results based on prior knowledge. Furthermore, in combining the translocation data with protein copy number estimations, we provide a genuine systems-biology approach to EGF signalling at the protein level (*Figure 5*). Unlike transcriptomic or proteomic profiling, our approach allows detection of cellular rearrangements at very early time points after stimulation, long before changes in protein abundance occur. The entire experiment (triplicate comparisons, six maps) required only five days of mass spectrometry measuring time.

In total, our analysis identified 40 translocation events, including numerous previously unreported movements. Among them are ten major regulators of actin dynamics, such as the kinases ROCK2, PKN2, PIK3C2B, and their downstream targets ADD1 and CTNN1, as well as PALLD, LASP1, and UTRN, suggesting that re-arrangement of the cytoskeleton is one of the major immediate effects of EGF signalling in HeLa cells. For several of these proteins, this study provides the first experimental evidence that they are targets of the EGF pathway (*Supplementary file 7*). Our data also reveal an unexpected cross-talk with other signalling pathways; Vasorin-shedding, AHNK and PDCD4 rerouting are all likely to counteract anti-proliferative TGFB signalling, and may serve to enhance EGF activity. Strikingly, we observed several transcriptional regulators leaving the nucleus. While nuclear import of proteins, such as ERK2/MAPK1, is a common downstream effect of signalling, nuclear protein export has been reported comparatively rarely. A possible explanation is that this type of movement is more difficult to detect with conventional approaches, such as microscopy: protein import concentrates the signal in the nucleus, whereas export diffuses it. Taken together, these observations highlight the power of the holistic proteomic approach, which identifies the co-ordinated behaviour of functionally linked groups of proteins, and thus uncovers cellular response modules.

## Outlook

This study demonstrates that dynamic organellar maps can shed new light even on relatively well-studied processes, such as EGF uptake. We propose that they will be similarly suitable in the fields of autophagy, membrane trafficking and cellular differentiation, providing a powerful complement to imaging-based techniques. Since they offer an unbiased approach to studying cellular dynamics

that does not require prior knowledge, they will also be an effective tool for exploratory investigations of poorly characterized processes. The possibility to combine maps with high-throughput phosphoproteomics data (*Humphrey et al., 2015*) promises to provide unprecedented views of signalling, by linking the movement of substrates to their phosphorylation status. Moreover, as we have shown here, organellar maps will pave the way for quantitative process modelling in cell biology.

## Materials and methods

*How to use the website www.mapofthecell.org*

## Quick start guide - from cells to organellar maps in 6 easy steps

The following is an overview protocol for the rapid generation of organellar maps, focusing on essential steps, and including an approximate time frame. Detailed descriptions may be found in the corresponding sections below.

| Step | Description | Time requirements |
|------|-------------|-------------------|
| | Starting material: SILAC heavy and light labelled HeLa cells (1 x15 cm dish each, 50% confluent, ie 2 x 10 million cells). | |
| 1 | Mechanical cell lysis, and differential centrifugation subcellular fractionation → the actual 'Fractionation Profiling' | 4 hr |
| 2 | Protein assay of fractions, overnight tryptic digest, peptide clean-up | 4 hr hands-on, + overnight digestion |
| 3 | Mass spectrometry analysis (Thermo Q-Exactive HF) | 20 hr (fast protocol) |
| 4 | MaxQuant data processing (free software), data filtering | < 24 hr (processor with 8 cores e.g. intel i7) |
| 5 | Visualization of maps by PCA, check clustering (using eg SIMCA software (free demo), or Perseus software (free) | 1 hr |
| 6 | Prediction of protein subcellular localization by SVM classification (Perseus, free software) | 1 hr |
| →From cells to map in 3 days | | |

*Visit www.MapOfTheCell.org for interactive exploration of the data provided in this study.*

## Experimental protocols

### Cell culture

HeLaM cells (*Tiwari et al., 1987*) were cultured at 37°C under 5% $CO_2$, in Dulbecco's Modified Eagle's Medium (DMEM) without Arginine, Glutamine, Lysine or Sodium Pyruvate (Gibco, #A14431-01), supplemented with 10% (vol/vol) dialyzed foetal calf serum (PAA, #A11-107), 1 mM Sodium Pyruvate (Sigma, #58636), 1 x GlutaMAX (Gibco, #35050-061) and, either; 42 mg/L $^{13}C_6,^{15}N_4$-L-Arginine HCl (Silantes, #201604302) together with 73 mg/L $^{13}C_6,^{15}N_2$-L-Lysine HCl (Silantes, #211604302), or 42 mg/L Arginine HCl and 73 mg/L Lysine HCl with standard isotopic constituents (Sigma, #A6969 and #L8662). Cells were passaged >7 times in these media before experiments began.

### EGF treatment

SILAC labelled HeLa cells were grown in 15 cm cell culture dishes containing 25 mL of medium, to 50–80% confluency. Recombinant Epidermal Growth Factor (Sigma, #E9644) was reconstituted in PBS (Gibco, #14190–094) and added to dishes at a final concentration of 20 ng/mL. Dishes were returned to 37°C for 20 min.

### Imaging of EGF internalization by fluorescence microscopy

HeLa cells were grown in 35 mm cell culture dishes with embedded glass coverslips (#P35G-1.5-14-C; MatTek, Ashland, US), to approximately 30–50% confluency. The growth medium was exchanged

for DMEM supplemented with 5 mM HEPES (Gibco, #15630–056), and 25 nM LysoTracker Red DND-99 (ThermoFisher Scientific, #L-7528, from 1 mM stock in DMSO), and cells were incubated at 37°C for approximately 20 min. Cells were then chilled on ice for several minutes. The growth medium was replaced with pre-chilled DMEM/HEPES supplemented with fluorescently labelled EGF (biotinylated, complexed to Alexa Fluor 488 Streptavidin; ThermoFisher Scientific, #E13345) at 2 μg/ml, as recommended by the manufacturer. Cells were incubated on ice for 30 min to pre-bind the EGF to plasma membrane EGF receptors. The medium was then replaced with cold DMEM/HEPES. One batch of cells (control) was fixed immediately, with 3% formaldehyde in PBS. The other batch was incubated at 37°C, for 30 min, to allow EGF uptake. Cells were fixed with formaldehyde, as above.

Microscopy was performed at the Imaging Facility of Max Planck Institute of Biochemistry, Martinsried, using a ZEISS (Jena, Germany) LSM780 confocal laser scanning microscope equipped with a ZEISS Plan-APO 63x/NA1.46 oil immersion objective. ZEISS Zen software was used to acquire images, and Adobe Photoshop CS6 was used for cropping and global brightness/contrast adjustments.

## Cell fractionation for generating organellar maps
### Scale
For generation of an organellar map, we recommend using two 15 cm (50–100% confluent) cell culture plates (1 x SILAC light, 1 x SILAC heavy, $\geq$10 million cells each). For HeLa cells, this typically yields >100 μg of protein in the subcellular fraction with the lowest yield (usually the 24K pellet), sufficient for several downstream mass spectrometric analyses. The preparation can be scaled down further if necessary (eg to 2 x 5 million cells).

### Harvesting cells and cell lysis
All steps were performed on ice with pre-chilled ice cold buffers. Cells were washed once with PBS (without $CaCl_2$ and $MgCl_2$; Gibco, 14190–094), and incubated with PBS for 5 min. PBS was removed and cells were washed once with hypotonic lysis buffer (25 mM Tris-HCl, pH 7.5, 50 mM Sucrose, 0.5 mM $MgCl_2$, 0.2 mM EGTA) prior to 5 min incubation in hypotonic lysis buffer. Plates were then drained of excess buffer by standing vertically for 1 min. Cells were scraped in 4 ml of fresh hypotonic lysis buffer, using a cell scraper (Sarstedt, #83.1831). Cells were transferred to a Dounce homogenizer (Sartorius, #8530700) and homogenized with 15 strokes with the tight pestle (Sartorius, #8530807). Following homogenization, sucrose concentration was immediately restored to 250 mM with hypertonic sucrose buffer (2.5 M sucrose, 25 mM Tris pH 7.5, 0.5 mM $MgCl_2$, 0.2 mM EGTA).

### Fractionation through differential pelleting
Crude cell lysate was transferred to a 15 mL tube and centrifuged at 1000 g for 10 min (Multifuge 1L, Heraeus), generating a pellet strongly enriched in nuclear material. Post-nuclear supernatant was transferred to a new 15 mL tube and centrifuged at 3000 g for 10 min (SILAC light lysate), or transferred to an ultracentrifuge tube (SILAC heavy lysate) and set aside. The SILAC light post-3000 g supernatant was transferred to an ultracentrifuge tube and centrifuged at 10,000 rpm (5400 g $RCF_{max}$) for 15 min (Optima MAX Ultracentrifuge, Beckman Coulter) using a TLA-110 rotor (Beckman Coulter). This process was repeated with centrifugation at 15,000 rpm (12,200 g $RCF_{max}$) for 20 min, and 21,000 rpm (24,000 g $RCF_{max}$) for 20 min. In the final spin, both the post-24,000 g SILAC light supernatant and the post-1000 g SILAC heavy supernatant were spun at 38,000 rpm (78,400 g $RCF_{max}$) for 30 min. The SILAC heavy organellar pellet was used as the reference organellar fraction and the supernatant as the cytosolic fraction. All pellets were resuspended in SDS buffer (2.5% SDS, 50 mM Tris pH 8.1) and heated for 5 min at 72°C.

### Protein concentration determination
Protein concentrations were determined using a micro-BCA assay (Thermo, #23225). Protein concentration standards of 0, 25, 125, 250, 500, 750, 1000, and 1500 μg/mL were prepared using a stock of 2000 μg/mL BSA (Thermo, #23209) and SDS buffer. Reagent A (containing Bicinchoninic Acid solution) was mixed 50:1 with reagent B (containing Copper(II)Sulphate), and 200 μL were plated into wells of a 96-well plate. 5 μL of standard solutions and samples of unknown concentration were

plated in triplicate. Plates were incubated for 35 min at 37°C before measuring the fluorescence following excitation at 595 nM using a plate reader (Infinite M200, Tecan). Sample concentrations were calculated by comparison to a standard curve.

## In-solution digestion of proteins

30 µg of each SILAC light-labelled fraction were independently mixed with an equal amount of SILAC heavy reference organellar fraction. Protein was precipitated by addition of 5 volumes of ice-cold acetone, before incubation at −20°C for 30 min and subsequent centrifugation at 4°C for 5 min at 10,000 g (Centrifuge 5415R, Eppendorf). In case of nuclear, organellar, and cytosol fractions, 60 µg of SILAC heavy labelled sample only were subjected to the same precipitation regime. All subsequent steps were performed at room temperature. Supernatants were removed and pellets allowed to air-dry for 5 min. Pellets were re-suspended in digestion buffer (50 mM Tris pH 8.1, 8 M Urea, 1 mM DTT) and incubated for 15 min. Cysteines were alkylated by addition of 5 mM Iodoacetamide with incubation for 20 min. Proteins were enzymatically digested by addition of 1 µg LysC per 50 µg of protein, with incubation for 3 hr. Digests were then diluted four-fold with 50 mM Tris pH 8.1 before addition of 1 µg Trypsin per 50 µg of protein and overnight incubation.

## Peptide purification and fractionation

Peptides were fractionated as previously described (*Kulak et al., 2014*). Briefly, SDB-RPS (Sigma, #66886-U) stage tips were activated with 100% Acetonitrile, followed by an aqueous solvent containing 30% (v/v) Methanol and 1% (v/v) TFA. Peptide mixtures were acidified with 1% (v/v) TFA, 15 µg (for single shot) or 25 µg (for fractionation) was loaded onto activated stage-tips, and washed with 0.1% (v/v) TFA. Peptides were eluted with 60 µL buffer X (80% Acetonitrile, 5% Ammonium Hydroxide) for single shot analyses (the 'fast maps' protocol). For fractionation (the 'deep maps' protocol), peptides were eluted using 20 µL SDB-RPSx1 (100 mM Ammonium formate, 40% (v/v) Acetonitrile, 0.5% (v/v) Formic acid), then 20 µL SDB-RPSx2 (150 mM Ammonium formate, 60% (v/v) Acetonitrile 0.5% (v/v) Formic acid), then 30 µL buffer X. Tryptic peptides were dried almost to completion in a centrifugal vacuum concentrator (Concentrator 5301, Eppendorf), and volumes were adjusted to 10 µL with buffer A* (0.1% (v/v) TFA, 2% (v/v) Acetonitrile).

## Mass spectrometry

4µL of peptides were loaded on a 50-cm column with 75-µm inner diameter, packed in-house with 1.8-µm C18 particles (Dr Maisch GmbH, Germany). Reverse phase chromatography was performed using the Thermo EASY-nLC 1000 with a binary buffer system consisting of 0.1% formic acid (buffer A) and 80% acetonitrile in 0.1% formic acid (buffer B). The peptides were separated by a linear gradient of buffer B from 2 to 30% in 130 min followed by washout (ramping to 95% B in 5 min, constant at 95% B for 5 min, ramping down to 2% B in 5 min, constant at 2% B for 5 min). The flow rate was 250 nl/min. The column was operated at a constant temperature of 55°C. The LC was coupled to a Q Exactive HF Hybrid Quadrupole-Orbitrap mass spectrometer (*Scheltema et al., 2014*) (Thermo Fisher Scientific, Germany) via the nanoelectrospray source (Thermo Fisher Scientifc, Germany). MS data were acquired using a data-dependent top-15 method, dynamically choosing the most abundant not-yet-sequenced precursor ions from the survey scans (maximum injection time 25 ms, 300–1650 Th). Sequencing was performed via higher energy collisional dissociation fragmentation with a target value of 1e5 ions determined with predictive automatic gain control. Isolation of precursors was performed with a window of 1.4 Th. Survey scans were acquired at a resolution of 120,000. Resolution for HCD spectra was set to 15,000 with maximum ion injection time of 55 ms. The 'underfill ratio,' specifying the minimum percentage of the target ion value likely to be reached at the maximum fill time, was defined as 20%. Repeat sequencing of peptides was kept to a minimum by dynamic exclusion of the sequenced peptides for 30 s.

## Bioinformatic analysis

### Overview

In this study, two independent sets of maps were generated:

1. Six replicate maps from untreated HeLa cells, to establish the method and to generate a database of protein subcellular localisation. These maps are referred to as the 'Static Maps' here.

2. Three untreated (control) and three EGF-treated (+EGF) maps, as part of a comparative experiment to demonstrate the applicability of the method to detect organellar translocation events. These six maps will be referred to as the 'Dynamic Maps' here.

Common procedures are listed without specific reference to either method. In some cases, processing of the Dynamic Maps required extra steps; these are described at the end of each section.

## Processing of mass spectrometry data

Mass spectrometry raw files were processed in MaxQuant (*Cox and Mann, 2008*) (v1.5.3.29), using the human SwissProt canonical and isoform protein database, retrieved from UniProt (www.uniprot.org). Raw files were organized into four parameter groups (PGs): double labelled SILAC data (PG1; all organellar subfractions); label-free (LFQ) data (PGs 2, 3, and 4; corresponding to single-labelled nuclear, organellar and cytosolic fractions, kept in separate PGs). Multiplicity was set to 2 for PG1, with Lys8 and Arg10 selected as heavy labels; Re-quantify was enabled for PG1; minimum number of quantification events was set to 1. Multiplicity was set to 1 for PGs2-4, with Lys8 and Arg10 selected as labels. LFQ was enabled for PGs 2–4; minimum number of LFQ peptides was set to 1; matching between runs (within PGs only) was activated, with a match time window size of 0.7 min, and an alignment time window of 20 min. Default parameters were used for all other settings.

## Mass spectrometry data filtering

The primary output from MaxQuant is the 'protein groups' file, which was used as the basis for all subsequent analyses. Identifications were filtered by removing matches to the reverse database, matches only identified by site, and common contaminants. The remaining identifications were split into three groups, for individual processing: Group 1, the SILAC ratio data (ie the subfractions obtained by differential centrifugation); Group 2, the intensity data for the nuclear, organellar and cytosolic fractions; and Group 3, the LFQ data for nuclear, organellar and cytosolic fractions.

Group 1 data were used to derive organellar maps. The raw data for one map are five SILAC ratios (called 3 K, 5.4 K, 12.2 K, 24 K, and 78.4 K, referring to the centrifugation speeds used to derive the corresponding fractions). For each map, we applied a stringency filter to ensure high accuracy of quantification, which increases with the number of peptide quantification events. First, we only considered proteins with complete profiles (ie a set of five SILAC ratios); any proteins with missing values were rejected. Second, we only retained proteins with three or more quantifications in each subfraction. In addition, proteins with only 2 quantification events in one or more subfraction were retained, if their ratio variability for ratios obtained with 2 quantification events was below 30%.

Group 2 data were used to derive protein abundance in the organellar, nuclear and cytosolic ('ONC') fractions, as well as overall protein copy numbers and concentrations. The raw data for each ONC triplet were three intensity values. As a stringency filter, we included only those proteins which were identified with at least two MS/MS events (ie without matching between runs) in at least one of the subfractions. All fractions were normalized to the same total intensity. Fractions were then weighted according to protein yields. Since the ONC split was highly reproducible, the same set of weighting factors was used for all maps generated as part of one experiment (0.20 for organellar, 0.32 for nuclear, 0.48 for cytosolic fractions - see *Figure 1—figure supplement 1*). For each ONC triplet, weighted intensities were summed for each protein, to derive whole-cell relative intensities. These were used to normalize the individual fraction intensities, to obtain for each protein a distribution profile over organellar, nuclear and cytosolic fractions (with the three values from a triplet adding up to one). Furthermore, whole-cell relative intensities were later used to estimate protein copy numbers and concentrations (see below, proteomic ruler). Please note that we chose to perform this analysis on intensity data, not LFQ data, as the three subcellular fractions ONC have very different composition. The LFQ algorithm works optimally on fractions with similar composition, so intensity data are more appropriate here.

Group 3 data were used to investigate changes in organellar, nuclear and cytosolic fractions between experiments. Only proteins with at least one MS/MS event in at least one fraction were kept. Please note that in principle, the Group 2 data could have been used here; however, LFQ offers slightly higher accuracy, includes a normalization step, and is conveniently generated by Max-Quant. As the Group 3 data are solely used for comparing equivalent fractions (ie nuclear with

nuclear, cytosolic with cytosolic, organellar with organellar, see below, 'Global protein distribution changes'), LFQ is the preferred choice here.

Please note that Group 2 and 3 data were derived from (single-label) SILAC heavy fractions. This is a consequence of the chosen labelling strategy (*Figure 1A*); it would be equally possible to invert the labelling for the entire workflow (ie SILAC light reference, organellar, nuclear and cytosolic fractions; SILAC heavy organellar subfractions), without affecting the experiment or interpretation of the data.

## Mass spectrometry measuring time requirements and map depth

We have implemented and tested two formats for making organellar maps, which differ in the required MS measuring time and achieved depth of coverage.

|  | Fast Map | Deep Map |
|---|---|---|
| Number of subcellular fractions | 8 (5xSILAC, 3xLFQ) | 8 (5xSILAC, 3xLFQ) |
| Peptide fractionations | 1 | 3 |
| Measuring time/sample | 2 hr 30 min | 2 hr 30 min |
| Total measuring time per map | 20 hr | 60 hr |
| Proteins mapped (average) | 2800 | 4500 |
| Global protein profiles (average) | 6000 | 8000 |
| Prediction performance (on markers) | 93.4% | 94.7% |

The Static Maps shown in *Figure 2*, *Figure 2—figure supplements 1* and *2* were prepared using the 'deep' protocol; the Dynamic Maps shown in *Figure 4* and *Figure 4—figure supplement 2* were prepared using the 'fast' protocol.

## Organellar marker set generation

The quality of machine-learning based classification depends strongly on the markers available to train the algorithm. Generation of a reliable organellar marker set is not trivial, as the subcellular localization of many proteins has not been unambiguously determined. Protein localization is often cell type specific, or dependent on the physiological context, further complicating the selection. Moreover, marker sets need to be extensive, to allow accurate cluster delineation. To our knowledge, there is currently no published widely accepted canonical organellar marker set. Hence, we manually curated such a set, optimized for the HeLa cell line used in this study (*Supplementary file 1*; 1076 proteins). Our selection criteria for a suitable marker were as follows. A) The protein must be expressed in HeLa cells and be identifiable by proteomics. Hence, all markers were picked from the list of proteins identified in the course of this study. B) The protein must have a well-documented and, as far as possible, unambiguous/predominantly unimodal steady state localization. Annotation of subcellular localization was gleaned from the various annotations in UniProt (www.uniprot.org), as well as from the protein's primary literature. C) The proteins chosen to represent a particular organelle must, more or less, cluster with other markers of this organelle. The function of the markers is to provide a reductionist model of the structure underlying the data; the most important property of a good marker set is hence to provide clusters that are stable over all future replicate maps. To achieve this, we selected suitable markers from several pilot PCA maps (which were not included in this study). We subsequently found their clustering confirmed in the maps presented here. D) Since classification by support vector machines is a 'boundary' method, marker proteins near the edge of a cluster are particularly important for cluster discrimination. Hence, once a core set of markers for organelles was established, we specifically augmented the set by identifying suitable candidates (based on criteria a-c) in the cluster periphery. In many cases, we even chose markers that visibly 'interdigitate' neighbouring clusters in PCA maps. The aim was to obtain the best boundaries, ie the

ones that afford the best predictive value for further proteins, and to minimise 'unchartered' space in between clusters. The marker set we provide here should serve as an excellent basis for the generation of maps in other cell types, although some fine-tuning may be required.

## Choice of compartment classes

The choice of classes was influenced by multiple factors. First, well-established organelles (Plasma membrane, ER, Mitochondria, Golgi, Lysosomes, Endosomes, and Peroxisomes) were included. In addition, the maps themselves suggested deeper resolution in places; thus, known markers of the ERGIC formed a discrete cluster, which also contained many markers of the cis-Golgi. As cis-Golgi is formed from coalescing ERGIC, this is not surprising. Interestingly, the mid- and trans-Golgi markers formed a separate cluster, and hence we decided to define a combined ERGIC/cisGolgi compartment, separate from ER and Golgi. We also observed a small but compact cluster of proteins known to associate with high-curvature ER (REEPs, Atlastins, and Reticulons). This cluster's map position is distinct from the much larger main ER cluster. Its vicinity to the ERGIC cluster may suggest that this cluster corresponds mostly to tubular ER structures; but since high-curvature ER markers are also found at the edges of sheet ER, we decided to call this cluster 'ER, high curvature'.

There are several types of endosomal compartments, ranging from 'early', 'late', to multivesicular bodies (MVBs), and endolysosome (the hybrid compartment produced from fusion of MVBs and lysosomes). As lysosomes are then re-formed from endolysosomes, they share many components. Any definition of endosomal compartments through marker proteins is therefore somewhat arbitrary. In our maps, the endosomal and lysosomal clusters are adjacent. We observe a relatively clear division of established pure lysosomal markers (such as lysosomal hydrolases) from known endosomal markers (eg Retromer, Transferrin receptor). Within the endosomal cluster, typical late endosomal markers (ESCRT-0, vacuolar ATPase subunits) are more concentrated near the lysosome-proximal side. This difference in distribution between late and early endosomes is suggestive, but not to the point where it warrants separate clusters for SVM analysis.

Finally, we observed a clear division of membrane-bound organelles from large protein complexes (such as ribosomes, proteasomes, and many others, *Figure 2—figure supplement 1*). Many of these complexes are very well resolved. However, since the primary aim of this method is to classify organellar proteins, we considered all large protein complexes as one large cluster. The contents of the large protein complex cluster are further discussed below.

## Visualization of organellar maps through principal component analysis

Each organellar map consist of a matrix of protein IDs associated with five SILAC ratio columns (Group1 data, see above) indicating for each protein the relative enrichment and depletion over the five subcellular fractions. For compact visualization of the data, principal component analysis was performed to achieve dimensionality reduction, using SIMCA 14 software (Umetrics/MKS - a free trial version of this software is available from their website: www.umetrics.com). Ratios were first log transformed, then scaled to unit variance. For each map, the first three principle components were calculated. We found that the best visual separation of organellar clusters was achieved by scatter plotting the scores of PC1 vs PC3.

In addition to the six individual maps (with 5 SILAC ratios each) analysed in *Figure 2—figure supplement 2*, a combined set with proteins common to all six maps (and thus 30 SILAC ratios) was used for generating *Figure 2*, as a conceptual 'consensus' of the six individual maps. All maps can be interactively explored online (www.MapOfTheCell.org). Similarly, the Dynamic maps in *Figure 4b* show maps combining the common sets from the three control maps (left), and the three EGF treatment maps (right), and are thus based on 15 SILAC ratios each. Individual control and treatment maps (5 SILAC ratios each) are shown in in *Figure 4—figure supplement 2*.

## Organellar maps through support vector machine classification

For the rigorous assignment of proteins to organellar clusters, a support vector machine (SVM) - based machine learning approach was used. Briefly, SVMs are a boundary separation method, which allows non-linear separation of clusters in multidimensional space. First, optimal boundaries between organellar clusters are determined using the marker proteins, with cross-validation to prevent over-fitting. Non-marker proteins falling within the boundaries of a particular cluster are then assigned to

that organelle. Distance to the boundary reflects classification confidence (reviewed in *Varmuza and Filzmoser [2009]*).

Classification was performed on the fractionation profiles (SILAC data, Group 1, see above). Data were prepared as tab delimited text files, containing protein gene names and UniProt identifiers, as well as the five SILAC ratios constituting a single organellar map. Organellar marker proteins were annotated with their corresponding compartment (listed in *Supplementary file 1*). The analysis was performed in Perseus software (version 1.5.2.11; www.perseus-framework.org). The marker compartments were imported as categorical annotations. Data were first log-transformed. The support vector machine (SVM) algorithm was then trained on the marker proteins, to determine optimal classification parameters. A radial basis function (RBF) was used as the kernel. Parameters Sigma (curvature) and C (misclassification penalty) were optimized by systematic scanning, to achieve the minimal overall classification error. 16-fold cross-validation (CV) was used to prevent over-fitting. Classification was then performed with optimal settings on both marker and non-marker proteins. This time, classification performance for the marker proteins was gauged by full (leave-one-out) cross validation, as a reflection of prediction error for each compartment. For the six replicate Static Maps, the results of the full CV are shown in *Supplementary file 2*. The average classification performance was 94.7% correctly assigned markers. Optimal values for Sigma ranged from 0.22 to 0.48, and from 8 to 35 for C.

For each protein, the analysis provides a prediction score for each compartment. Positive scores indicate class membership (ie the protein falls within the boundaries of the compartment model derived from the corresponding marker proteins); several class memberships can be predicted for one protein. The class with the highest score is the top prediction (even if the top score is negative, ie the protein is outside the boundaries of the top prediction; in this case, the assignment is a nearest cluster 'best guess').

To combine the results of several SVM analyses into one joint prediction, we explored two approaches. First, we analysed the set of proteins common to all six static maps, using all (6 x 5) = 30 SILAC ratios to build an SVM model. The prediction performance on the marker set was better than that of each single map (97.5% correct); however, the depth of this 'common' map was substantially lower (3766 proteins vs 4500 in each single map). We therefore employed an alternative approach to combine maps. We trimmed the results from each individual SVM analysis to include only positive scores. Scores were then summed for each compartment and protein across the six different analyses. The top summed score represents a protein's top cumulative compartment prediction. The advantages of this approach are two-fold: firstly, the full information from all replicate maps is used, not just a reduced common set; and secondly, it focusses on clear positive classification decisions. Thus, if two clusters are well separated in one map, but poorly separated in another, only the information from the better resolved dataset will be propagated. In addition, the procedure takes into account the consistency and number of times with which a protein is classified (eg a protein predicted with a medium score in six replicate maps achieves a higher cumulative score than a protein with only a single strong classification). The prediction performance of this approach is very good (96.5% correctly predicted), clearly exceeding the performance of each of the six individual maps. Furthermore, while it has similar performance to the map combining the proteins common to all six maps, it increased the number of covered proteins by 40% (from 3766 to 5265). Therefore, the results of this 'additive' analysis were used for construction of the HeLa spatial proteome database (*Supplementary file 4*).

As the numerical values of SVM scores are not immediately interpretable, we scaled the scores to percentiles. We ranked the cumulative SVM scores from all 1038 correctly predicted organellar marker proteins. Each score was then converted into a percent-rank. The SVM scores of non-marker proteins were converted into percentile scores, using this scale. For example, a protein with percentile prediction score of 30 has a cumulative SVM score that is greater than 30% of the correctly classified marker proteins.

To further put the scores into perspective, we investigated map concordance (ie prediction agreement between replicate maps, see below) as a function of scores. When the predictions from two individual maps are compared, the likelihood of concordance increases with prediction score. We defined prediction confidence classes accordingly:

| Prediction Confidence | Score | Map Concordance | Maps disagree in: | Predictions in this class |
|---|---|---|---|---|
| Very low | <1 | 94% | 1:13 | 657 |
| Low | 1-4 | 96% | 1:25 | 573 |
| Medium | 4-16 | 98% | 1:50 | 999 |
| High | 16-52 | 99% | 1:100 | 1229 |
| Very High | 52-100 | >99% | <1:100 | 1807 |
| Total | 5265 | | | |

For example, for proteins classified in two maps with a score > 20, both maps will make the same predictions for 99% of them. The concordance values were taken from the pairwise comparison of predictions from the six static maps; we chose the values derived from the inter-day comparisons as the most pessimistic estimate (see below, and *Figure 2—figure supplement 2C*).

SVMs were also used to predict subcompartments for ER (lumen and membrane) and mitochondria (matrix, outer and inner membrane). Starting with the combined set of predictions from the six static maps, all marker proteins of ER and Mitochondria which had been correctly predicted were further annotated for subcompartment localization (provided by UniProt (www.uniprot.org) and primary literature). SVMs were then applied separately to ER and Mitochondrial markers to define boundaries for the subcompartments. After optimization of Sigma and C as above, full cross-validation was used to judge prediction accuracy. Combination of the output from the six static maps showed the following levels of prediction performance for subcompartments:

| Sub-Compartment | No of markers | Correctly predicted | % |
|---|---|---|---|
| ER | | | |
| Membrane | 78 | 72 | 92.3% |
| Lumen | 49 | 40 | 81.6% |
| Total | 127 | 112 | 88.1 |
| Mitochondria | | | |
| Matrix | 173 | 171 | 98.8% |
| Outer membrane | 20 | 15 | 75.0% |
| Inner membrane | 46 | 37 | 80.4% |
| Total | 239 | 223 | 93.3% |

SVMs were then applied to the predicted non-marker ER and mitochondrial proteins, to provide subcompartment predictions.

Processing of the six Dynamic Maps was performed likewise, but without subcompartment annotation. In addition, control maps and +EGF maps were treated separately, and combined into two prediction outputs, ie organellar predictions before and after EGF treatment. *Supplementary file 2* lists the results of the prediction accuracy with full cross-validation. Since the Dynamic Maps were prepared with the 'fast' mass spectrometry protocol, which requires only one third of the measuring time of the 'deep' protocol used for the Static Maps, their performance is slightly lower.

## Estimating overall prediction accuracy

There is no universally suitable and accepted set of organellar marker proteins (see above, Organellar marker set generation). The true prediction accuracy of the classification achieved in this study is hence difficult to gauge. Nevertheless, prediction accuracy of the training set, estimated with leave-one-out cross validation, can be used as a proxy. Thus, for a set of 1000 marker proteins, support-vector machine (SVM) models are built and predictions made 1000 times, using a different combination of 999 marker proteins each time. This essentially simulates the prediction of proteins that are not in the marker set, and ensures that the SVM models have maximum predictive value for non-maker proteins. The marker protein prediction accuracy hence provides an estimate of the overall

performance for those new predictions that fall within the boundaries defined by the markers. External validation of our predictions supports this notion: the overall concordance with the recent LOPIT study (*Christoforou et al., 2016*) is 91.6%; the concordance with experimentally validated mitochondrial proteins (*Calvo et al., 2016*) is 97%, and hence close to the prediction accuracy of our mitochondrial marker set (98.8%, *Table 1*). Prediction confidence varies between individual proteins (see above, Organellar maps through Support Vector Machine classification), and also varies between organelles. As a conservative estimate of global prediction accuracy achieved in this study, we consider the average accuracy per membrane bound organelle, 92.7% (*Table 1*).

## Map concordance analysis

To estimate the reproducibility of our organellar maps, we determined map concordance, defined as the percentage of identical predictions between two independent map replicates. We compiled the individual top predictions and SVM scores for the six replicate Static Maps into a table (with 12 classified compartments). We then performed 15 pairwise comparisons (all combinations of the six maps), including 12 inter-day, and 3 intra-day comparisons (maps 1 vs 2, 3 vs 4, and 5 vs 6). Concordance was calculated as follows. First, the set of proteins present in both maps was determined. Next, the proportion of proteins with identical top compartment predictions was calculated, corresponding to the map concordance. The average concordance across all inter- and intra-day comparisons was then determined. As concordance is a function of prediction confidence, we next introduced SVM scores as a further criterion. Scores for individual maps were converted to approximate single map percentile scores, by dividing the summed percentile scores (see above) by 6. Average map concordances were then calculated after filtering for predictions present in both maps with a given minimum score.

Furthermore, we compared the combined output from the six Static Maps prepared with the 'deep' mass spectrometry protocol (60 hr) with the combined output from the three controls of the Dynamics maps, which had been prepared with the 'fast' protocol (20 hr). Map concordance between these two sets was calculated as a function of prediction scores. Concordance was very similar to that observed for intra-experimental comparisons of single static maps (*Figure 3*), at 98% with a score cut-off at 7, demonstrating a high level of agreement between maps generated with the 'deep' and 'fast' protocols.

## Intensity data analysis of global protein distribution

The intensity data for nuclear, organellar and cytosolic fractions (Group 2 data, see above) were used to evaluate a protein's global cellular distribution. It is very important to separate this analysis from the organellar assignment achieved via the SILAC based maps. Many proteins have a cytosolic pool as well as a (peripheral) membrane associated pool. If the cytosolic fraction was added to the organellar maps, it would show these proteins more or less near the cluster of purely cytosolic proteins, but not reveal the organellar association of their membrane associated pool. Hence, we decided on a two-tiered classification, which first quantifies a protein's global distribution, and then reveals its membrane-bound compartment. The same logic applies for the numerous proteins that shuttle between nucleus and cytosol. Our intensity analysis identifies which proteins are predominantly nuclear or predominantly cytosolic, and which proteins are likely to shuttle between the two (pronounced nuclear and cytosolic pools). Again, inclusion of these data with the organellar maps would not allow the identification of nuclear/cytosolic shuttling proteins.

The intensity analysis shows for each protein a distribution across nuclear, cytosolic and organellar fractions, normalized to 1; profiles from different proteins are thus directly comparable. We broadly classified proteins based on the relative contributions of each pool, using useful arbitrary cut-offs:

| Class | Abbr. | Nuclear Pool | Cytosolic Pool | Organellar Pool |
|---|---|---|---|---|
| Mostly nuclear | N | $\geq 0.85$ | | |
| Mostly cytosolic | C | | $\geq 0.85$ | |

| | | | | |
|---|---|---|---|---|
| Mostly organellar* | O | | <0.15 | >0.3 |
| Organellar/ cytosolic | O/C | <0.4 | >0.15 | >0.15 |
| Nuclear/ cytosolic | N/C | >0.15 | ≥ 0.15 | <0.1 |
| Broad distribution | B | Any proteins not classified above | | |

*Please see below "Interpretation of global intensity distribution profiles', why these cut-offs are chosen differently.

This classification scheme is also incorporated into the HeLa spatial proteome database (*Supplementary file 4*).

## Interpretation of global intensity distribution profiles and their changes

The global intensity distributions must be interpreted in light of the subfractionation scheme. Cells are lysed mechanically, centrifuged at low speed to obtain the nuclear fraction, and at high speed to obtain the organellar fraction. The supernatant is the cytosolic fraction (see above for details). The nuclear fraction contains essentially all nuclei; this is supported by the fact that we detect over 400 proteins exclusively (>99%) in this fraction, including abundant ones. However, the nuclear fraction is fairly crude, and contains burst or partially lysed cells, as well as some mitochondria; based on the proteomics data, we estimate that it contains ~60% nuclear protein, and 40% protein from other membranes. As a consequence, genuine organellar proteins tend to be recovered partly in the nuclear fraction, and partly in the organellar fraction. The distribution over the two organelles depends on the compartment. For example, lysosomes are released very effectively from cells by our lysis conditions; lumenal lysosomal proteins are typically recovered >90% in the organellar fraction. In contrast, since partially lysed cells will also pellet with the nuclear fraction, and retain large plasma membrane sheets, plasma membrane proteins are typically only recovered at around 45% in the organellar fraction. Most other organelles fall in between these two extremes. The interpretation of the resulting abundance profiles is hence straightforward in some, and more complex in other cases.

1. If a protein has a very large (>85%) cytosolic pool, it is predominantly cytosolic.
2. If a protein has a large (>85%) nuclear pool, it is predominantly nuclear.
3. Some proteins have both substantial cytosolic and nuclear pools (>15% each), which when combined account for >90% of the protein's distribution. Such proteins are nuclear and cytosolic, as is typical of many nuclear shuttling proteins.
4. The protein has a very substantial (>30%) organellar pool, but a small (<10%) cytosolic pool. This is typical of (integral membrane or lumenal) organellar proteins. The proportion of material in the organellar fraction depends on the compartment, and varies between 30% and 100%. The apparent 'nuclear' pool is in most cases a consequence of poor organellar release.
5. The protein has both substantial organellar and cytosolic pools (>15% each); when combined, they account for >60% of the protein's distribution. This is typical of peripheral membrane proteins that have a cytosolic pool, as well as a membrane associated pool.
6. Some proteins have a 'broad' distribution with pools in all fractions; many very abundant proteins fall into this category.
7. The above classification cannot easily identify those proteins that are genuinely both organellar and nuclear.
8. Many protein complexes are sufficiently large to pellet under our fractionation conditions (eg ribosomes and proteasomes). Large protein particles thus appear to have an 'organellar' pool; the proportion depends on the size of the complex, and also if it can associate with organellar membranes. For example, the proteasome has a cytosolic pool >90%, the ribosome only ~35%. For proteins classified as 'large protein complex', the abundance profiles hence reflect a combination of factors, which must be evaluated case by case.

A simplified look at the profiles compares only the cytosolic vs the non-cytosolic pool (ie organellar plus nuclear fractions). This principally predicts to what extent a protein is associated with membranes (organelles or the nucleus). The HeLa spatial proteome database (*Supplementary file 4*) shows both types of profile.

## Why use a crude nuclear fraction?

Please note that the nuclear fraction was deliberately not 'purified', as in traditional fractionation approaches. The guiding principle for our preparation was to retain all cellular material, to allow global protein quantification/copy number determination, and also to maximize speed of preparation (to prevent organellar deterioration and leakage). Any further purification of the nuclear fraction would by definition have caused the loss of material, both nuclear and organellar, and thus distorted any attempt of quantification. Nuclear leakage through longer preparation times would have further invalidated the analysis.

## Copy number estimation

Absolute protein copy numbers were calculated using the 'Proteomic Ruler' approach (*Wisniewski et al., 2014*). Briefly, this method quantifies the mass spectrometric signal of histones, which is proportional to cellular DNA content. Knowledge of the cell's chromosomal ploidy in turn allows the conversion of the histone signal into histone protein mass per cell. The relative intensities of all proteins can then be converted into copy numbers per cell, as well as cellular concentrations. The proteomic ruler approach is remarkably accurate; comparison to 'gold standard' quantification with isotope labelled peptides (PrEST approach [*Zeiler et al., 2012*]) shows similar precision, and typical absolute deviations of less than two-fold (*Wisniewski et al., 2014*).

The proteomic ruler has been implemented as a convenient plug-in for the Perseus platform (www.perseus-framework.org). To derive protein copy numbers, whole-cell protein intensities (ie pooled and weighted Group 2 data, see above) were imported into Perseus 1.5.1.16. For the Static Maps, we analysed six biological replicates. For the Dynamic Maps, three replicates for control and three for EGF treatment were analysed (separately from the Static Maps). Signals were normalized to average molecular mass. Ploidy was set to three (a reasonable estimate for HeLa cells –the exact ploidy is not known and may vary). Normalization was the same across all columns. A total cellular protein content of 200 g/l was assumed (default). Medians and standard deviations were calculated from repeat experiments to provide copy number, concentrations, and error estimates.

To assess the quality of our analysis, we first compared the prediction consistency from the individual Static Maps (six biological replicates). The average Pearson correlation of copy numbers between replicates was 0.98 (calculated from 15 pairwise comparisons, log data), demonstrating very high precision (ie prediction reproducibility). To gauge prediction accuracy, we compared median copy numbers predicted from our maps with absolute copy numbers of 27 proteins derived in an independent study using the PrEST approach (*Zeiler et al., 2012*). Pearson correlation of the two datasets (in log space) was 0.96, and thus remarkably high, showing a near-linear relationship of our data with the PrEST data. The median ratio of copy numbers (static maps/PrEST) was 2.3, suggesting a small systematic shift in our predictions relative to the PrEST data. The two studies used different strains of HeLa cells (which may differ in cell size), and this may account for the shift. In any case, the proposed overall accuracy of the proteomic ruler is also in that range (*Wisniewski et al., 2014*). In sum, the copy number data obtained here show excellent internal consistency, and are in good agreement with independently obtained data.

The proteomic ruler approach was also used to estimate copy number changes in the Dynamic EGF treatment experiments. Copy numbers were log-transformed, and medians were calculated from the three control and the three EGF treatment maps. In addition, for each protein the copy number difference between control and treatment maps was analysed with a paired t-test (two-tailed). A p-value <0.01 was considered as significant (no correction for multiple hypothesis testing was applied, to maintain statistical power; in this context, false positives are not too detrimental, as the actual movement of proteins between compartments is assessed in separate statistical tests (with full FDR control). For proteins with no significant change, the overall median from all six maps (three controls and three treatment maps) was used as the basis for further calculations. For proteins with significant changes, the copy numbers for control and treatment were kept separate for subsequent calculations.

Copy numbers were then multiplied with the proportions of nuclear, organellar and cytosolic fractions (see above, intensity analysis). Average differences between corresponding fractions before and after treatment were calculated. For proteins with insignificant overall copy number changes, all compartment-specific changes balance to a net overall change of zero.

## Organellar leakage analysis

Organellar integrity is an important concern for cell lysis conditions. We assessed preparation-induced leakage of mitochondria, lysosomes, and ER, by analysing the cytosolic pool of known lumenal marker proteins of these organelles (see *Supplementary file 1*) for the six Static Maps (*Figure 1—figure supplement 1*). The underlying assumption is that these proteins should have almost no cytosolic pool if organelles stayed intact during cell lysis; conversely, any cytosolic appearance of markers should indicate the magnitude of leakage. In all three cases, we observed a large proportion of proteins with no cytosolic pool (<1%); a distribution of proteins with small cytosolic pools (1–5% for ER and mitochondria, and 1–15% for ER); and a small number of proteins with large cytosolic pools (>20%). We interpret these three groups as follows. 1) Some lumenal proteins are attached to the membrane of the compartment, and will not leak if the compartment is damaged. 2) Genuine protein leakage will vary from protein to protein, possibly with size, solubility, binding partners, etc. 3) Some proteins have splice variants with cytosolic localization, giving rise to apparent large cytosolic pools even in the absence of leakage (and this is in fact documented for many of the proteins we observe in group 3). Only the second group of proteins is hence likely to reflect genuine leakage. We therefore calculated the averages from proteins with cytosolic pools between 1–20% only, indicating leakage of 3.9% for mitochondria, 2.3% for lysosomes, and 8.3% for ER. These values suggest a very high level of organellar integrity achieved by our lysis procedure. The slightly higher value for ER is also expected, as the reticular morphology and interconnectedness with the nuclear membrane makes it impossible to extract this organelle intact.

## Generation of the HeLa spatial proteome database

The HeLa spatial proteome database (*Supplementary file 4*) is an interactive Excel-file that allows users to enter query proteins and retrieve predictions for subcellular localization, relative protein abundance, absolute copy number and concentration, as well as the distribution over organellar, cytosolic and nuclear fractions. In addition, a 'neighbourhood analysis' is performed, revealing which proteins share the query protein's fractionation behaviour. Using the database is quick and easy; you only need to enter the gene name or UniProt ID of the protein(s) of interest. The database automatically retrieves all associated information. Multiple simultaneous queries can also be submitted as a list of gene names/UniProt IDs ('batch submission). The database includes a quick start user's guide, as well as guidelines for interpreting the output.

The database construction is straightforward. Mostly, it retrieves information from a central data table, prepared and compiled as described in the various sections of the Methods. An exception is the neighbourhood analysis, which is calculated within the database. Briefly, the squared Euclidian distance to the query protein's profile is calculated for every protein in the database. Proteins are then sorted by ascending distance to the query. Those with low distances have fractionation profiles similar to the query. In many cases, this identifies members of stable protein complexes, but may also reveal other important/interesting connections. Useful distance cut-offs to guide the interpretation of profile similarity were derived by analysing the distances of between members of known stable complexes, as described (*Borner et al., 2014*).

The Dynamic EGF spatial proteome database (*Supplementary file 6*) is similar to *Supplementary file 4*, but includes information on both control and EGF treatment maps. As an additional feature, it highlights differences in protein compartment prediction, distribution, as well as copy number, before and after EGF treatment.

## Z-filtering to enhance the prediction of nuclear pore complex and actin binding proteins

*Supplementary file 4* and *Supplementary file 6* also include a further quality filter for proteins predicted as part of the nuclear pore complex (NPC) and the actin binding proteins (ABP). Markers for these two clusters have characteristic, very low ratios of organellar to nuclear pools (see above), and this can be used as a further distinguishing filter. First, the organellar/nuclear (ONB) balance was calculated for all proteins predicted as NPC or ABP. For both compartments, the median ONB and a robust standard deviation (1.483 * median absolute deviation from the median) were calculated in $\log_2$ space for known marker proteins only. For each other protein predicted as NPC or ABP, the $\log_2$ (ONB) was subtracted from the compartment's median, and divided by the robust standard

deviation (z-transformation). For NPC, predictions with z-scores >5 were rejected, and for ABP, predictions with z-scores >4. These proteins were then re-assigned to their second highest SVM classification (if available). In total, this led to the re-assignment of 61 proteins.

## Protein mass analysis of subcellular compartments

Calculating the mass of each subcellular compartment required the 'Intensity data analysis of global protein distribution' (Group 2 data), copy number estimations, and the organellar classifications. Initially the median copy numbers were multiplied with the MW (kDA) to get the mass of each protein in the cell. These masses were then multiplied by the average proportion of the intensity from the three fractions; Organellar, Nuclear and Cytosolic. This gives a mass for each protein in each of these three compartments. However, the contamination of the nuclear fraction with non-nuclear organellar material would lead to incorrect assignment of mass to the nucleus. We therefore used the 6 classifications described in 'Intensity data analysis of global protein distribution' to redistribute this misassigned nuclear mass as follows:

| Global class | Cytosolic Fraction | Nuclear Fraction | Organellar |
|---|---|---|---|
| Mostly Nuclear | C | N+O | - |
| Most cytosolic | C | N | O |
| Mostly organellar | C | - | O+N |
| Organellar/ Cytosolic | C | - | O+N |
| Nuclear/ Cytosolic | C | N+O | - |
| Broad | C | N | O |

For example, for a protein that is mostly organellar, its calculated cytosolic pool will be as determined; its organellar pool will be the sum of its nuclear and organellar pools.

Several assumptions were made in order to define these re-distributions.

The first assumption is that everything that is found in the cytosol genuinely has a cytosolic pool. Although there will be some protein amount coming from the nucleus and leakage from organelles, it is not possible to determine which nuclear proteins genuinely have a cytosolic pool and which are contaminants due to leakage, therefore the mass of the cytosol is calculated as everything detected in the cytosolic fraction.

The second assumption is that nuclear proteins found in the organellar fraction are contaminants from the fractionation procedure. Not all nuclear material will pellet at 1 K, especially broken nuclei. Therefore the mass of the organellar fraction of proteins defined as nuclear is given to the nucleus. In addition, Nuclear/Cytosolic proteins have their organellar pool given to the nucleus for the same reason.

Conversely, organellar proteins found in the nuclear fraction are also deemed contaminants, this is particularly important since many organelles are not efficiently released upon cell lysis, due to attachment to the cytoskeleton, cells remaining partially intact, incomplete stripping of membranes that are physically attached to the nucleus, such as the ER. Therefore any protein that is classified as organellar is assigned both the organellar and nuclear pool of that protein. Likewise, for organellar/ cytosolic proteins.

Proteins that are broadly distributed are assumed to be genuinely broadly distributed and are assigned to the nucleus and organellar fraction in the proportions determined in the intensity analysis.

Proteins classed as cytosolic have their organellar or nuclear mass assigned to the nucleus or the organelle fractions.

The mass of the organellar fraction can then be divided to the respective organelles using the organellar classification determined by SVM analysis of the organellar maps. From this classification,

large protein complex proteins were assigned to cytosol, because these mostly pellet due to their mass, and not necessarily because of membrane association, as shown by their enrichment in the 80K fraction, where all membranous organelles are de-enriched.

Any protein assigned to Actin-binding protein cluster in the organellar maps, regardless of its split among organellar, nucleus or cytosol, is considered as part of the cytoskeleton. The cytosolic pool of actin binding proteins may truly reflect a soluble pool, hence the mass of Organellar and Nuclear pools constitute the actin cytoskeleton and the Cytosolic pool represents the soluble actin cytoskeleton components.

## Detection and interpretation of translocation events – dynamic organellar maps

### Overview
Translocation events are detected at two levels: 1) changes in organellar association; 2) changes in global profile distribution (organellar, nuclear, cytosolic). Each analysis is also a two-step procedure: First, a statistical test is used to identify reproducible translocations. Next, the nature of the translocation is evaluated in the context of all available spatial information (eg comparing organellar predictions, PCA maps, cytosolic vs membrane distributions, copy number changes, etc.). The rationale for decoupling detection and interpretation of movement is explained at the end of this section.

### Detection of dynamic changes between organellar maps
We have developed a statistical test for the detection of protein movement between organellar maps. The test combines two simple metrics, protein translocation magnitude (M) and translocation reproducibility (R), which can be combined into an easily interpreted 'MR Plot'. The test is optimally performed on three biological repeats of control and treatment each, as demonstrated in this study. It is still useful for analyses with two repeats, but less stringent. If a comparative experiment is performed only once (eg as a pilot experiment), only the magnitude of translocation metric can be calculated. This still indicates all relevant changes, but with a much higher background (ie false positive rate). The test can conceptually be divided into several data transformation steps, and the subsequent calculation of the two metrics. The following is a step-by-step guide, assuming a simple comparative experiment has been performed in triplicate (three treated and three control maps).

### Required data
A typical fractionation profiling map consists of a list of identified proteins associated with five SILAC ratios (the protein's profile). We routinely use stringent filters to include only high quality data in the analysis (see above, Group 1 data). Furthermore, for maximum stringency we only consider proteins that have complete profiles in all generated maps (ie in case of triplicate biological repeats of a map comparison, only proteins with 5 SILAC ratios in all six maps are included). It is not recommended to allow any missing values, or perform any kind of data imputation. Compile the data into a single table, with columns for gene name, unique protein identifier (UniProt ID), and all 30 SILAC ratios from the six maps (3 control, 3 treated). In addition, the relative average protein yields from the 5 SILAC fractions are required (expressed as five fractions summing to a total of 1). These will serve as weights for the individual fractions. As the yields tend to be very stable between maps (*Figure 1— figure supplement 1*), it is acceptable to use the same set of (average) yields for all maps.

### Data transformation
The following transformations are performed for each map individually. SILAC H/L ratios (default output from MaxQuant) are inverted (to L/H). Ratios now indicate the level of enrichment of a protein in each fraction relative to the reference fraction. Each of the five ratios is then multiplied by its corresponding fraction weight, to account for the different protein yields in the fractions. For each protein, the weighted ratios are summed across the five fractions. Each weighted ratio is then divided by the total for this protein. This results in a distribution profile with five values, which sum to 1. The ratio profiles have thus been converted into normalized profiles, reflecting for each protein the proportion pelleted in each fraction. Apply this transformation to each of the six maps.

## Calculation of delta profiles

The three biological replicates are divided into pairs of control and treatment maps. Typically, the control and treatment maps prepared on the same day are treated as a cognate pair. For each cognate pair, a Delta Matrix (ie the difference matrix between both maps) is calculated. This is achieved by simple subtraction of the normalized profiles, fraction by fraction. For each protein, a 'difference profile' of five values is calculated. The array of all these Delta profiles is the new Delta Matrix. An experiment with three replicates thus generates three Delta Matrices (with five data columns each). The delta profile reveals how much and in what way a protein's fractionation behaviour has changed between control and treatment.

## Magnitude of translocation (M) score

Proteins that do not translocate should have delta profiles close to zero (baseline profile; ie same fractionation behaviour before and after treatment). Proteins that undergo translocations should have delta profiles with large changes. To determine significant deviations from baseline changes, a multivariate outlier test is performed. The underlying assumption is that in a comparison of two identically prepared maps (eg control vs control), all delta profiles will reflect experimental noise only, and no significant changes are expected. Furthermore, this noise should be Gaussian, and the Delta Matrix hence approach a multivariate normal distribution. This should also largely be true for a comparison control vs treated, were only a few proteins show genuine movement. Consequently, the Delta Matrix should be multivariate normal, with the exception of the translocating profiles. To detect these multivariate outliers, the squared Mahalanobis distance to the center of the data is calculated. This follows a Chi-square distribution with n degrees of freedom (n = number of data points, ie five for the maps). The Mahalanobis distance requires calculation of the covariance matrix. Since this is strongly affected by outliers, a robust calculation is performed, using the minimum covariance determinant method (*Fauconnier and Haesbroeck, 2009*). Robust distances are then converted into p-Values (ie likelihood of expecting a distance as great, or greater, than the observed one, by chance). Multiple hypothesis testing correction is performed using the Benjamini-Hochberg approach.

Please note that summing all absolute values of a delta profile would be much less informative than the described outlier test, for two reasons: firstly, the outlier test takes into account correlations between changes; and secondly, the test takes into account the different variances of the delta values.

The outlier test has been implemented in Perseus ('Multidimensional Significance'; www.perseus-framework.org), in a user friendly format. For each cognate pair of maps, the Delta matrix (generated as a tab-delimited file in Excel, for example) can be uploaded. It is recommended to use 90% of the raw data for calculating the minimum covariance determinant (ie n*0.9). A p-value indicating translocation significance is calculated for each protein. Proteins are ranked from lowest to highest p-value, and a Q-value (estimated FDR cut-off) is calculated by multiplying each p-value by the total number of proteins in the set, and dividing by the protein's rank (Benjamini-Hochberg correction for multiple hypothesis testing). Proteins below a predetermined threshold (eg 0.01) have significantly different profiles in control vs treatment map. Please not that this test is very sensitive, owing to the robust design.

The analysis is performed separately for all three cognate map pairs. For each protein, three Q-values are obtained. Q-values are transformed to M-scores ('Magnitude of translocation') by calculating $-\log_{10}(Q) = M$. (Example: $M \geq 2$ corresponds to an FDR $\leq 0.01$). Proteins undergoing genuine translocation events should have large M scores in each experiment.

## Reproducibility of translocation (R) score

If a protein translocates specifically in every experiment, it will show the same direction of shift every time (even if the magnitude of translocation varies). As a consequence, Delta profiles from different experiments should be highly correlated. Each protein is associated with three Delta profiles. Pearson correlation of Delta profiles is calculated for all three possible combinations (Replicates 1 vs 2, 1 vs 3, 2 vs 3). Genuinely translocating proteins will have very high positive correlations in every combination (in our experience, generally >0.8). FDR derived cut-offs are derived as described below.

## Combining M and R scores into a 'MR plot'

Proteins undergoing genuine specific translocation events must score highly for Magnitude and Reproducibility of translocation. In an experiment performed in triplicate, three separate M and R scores are obtained. To combine these values, two methods are recommended:

a) High stringency scoring (as used in the present study)

For each protein, select from the triplicates the lowest M score (ie highest p Value), and the lowest R score (ie lowest correlation). Plot these two values for each protein. Genuine hits are located in the upper right quadrant of the plot.

b) Sensitive scoring

For each protein, calculate from the triplicates the median M and R scores. Plot as in a).

For experiments performed in duplicate, only two M scores and one R score are available; for experiments without replicates (eg pilots), only one M score is available. Hence, the stringency will be much reduced.

## Determining score cut-offs

Significance boundaries for the MR plots can be derived by performing a mock experiment with no expected true positives. This yields empirically derived FDR values, and is hence preferable over theoretically derived p-value cut-offs.

In the present study, we treated the six replicate 'Static Maps' as a mock experiment. We treated maps 1, 3, 5 as controls, and maps 2, 4, 6 as 'mock treated', to simulate a comparative experiment performed in triplicate. Stringent M and R scores were calculated as described above, and visualized in a MR plot (*Figure 4E*). Score boundaries for M and R divide the plot into four quadrants: not significant in either M or R (lower left quadrant), significant in M or R, and significant in both M and R (upper right quadrant). We then chose stringent boundary settings (M=2, R=0.9) that left the upper right quadrant empty in the mock experiment, corresponding to an empirical FDR of 0 for our EGF treatment experiment (*Figure 4F*). For an exploratory analysis, we determined lower cut-offs with an expected FDR of 10% (M=0.6, R=0.7).

Please note that if the mock experiment has different depth from the treatment experiment, the expected FDR may be scaled accordingly. In the present study, the mock experiment had much greater depth than the EGF treatment experiment (3766 vs 2237 proteins). For the FDR = 0 settings, scaling makes little difference, and was not required. For the FDR = 10% settings, scaling was performed as follows: applying the chosen M (0.6) and R (0.7) scores to the mock data (with depth 3766 proteins) shows 3 significant hits; scaling by the factor (2237/3766) predicts 1.8 hits in an equivalent mock experiment with depth 2237 proteins. In the EGF experiment (with depth 2237 proteins), the same cut-offs reveal 18 significant hits; 1.8 expected false positives out of 18 corresponds to an FDR of 10%.

More lenient cut-offs may be chosen, and FDRs >10% may be tolerable, based on the experimental design and research question. In particular, the M-score may be lowered, to include smaller translocations.

## Evaluating organellar translocations

Once significant outliers have been detected in the MR plots, the underlying translocation events may be evaluated through PCA based maps, SVM organellar predictions, or nearest neighbour analysis. We recommend a combination of all three approaches, as the information gleaned is complementary.

PCA maps show the movement graphically, and are a good starting point for orientation; they reveal the approximate compartment before and after the translocation, and also indicate if the translocation event is likely to be only partial. Maps also show the relative position of a protein within a cluster (central vs near the edge; in the latter case, it may be of note which other cluster is close). As an example from this study, the EGFR is very central to the plasma membrane cluster in untreated cells, but is on the edge of the endosomal cluster after EGF treatment, in close proximity to the lysosomal cluster, hinting perhaps at a late endosomal compartment, or beginning lysosomal localization. (Please note that the endolysosomal compartment may be seen as a continuous distribution of compartments, with early endosomes and lysosomes marking the extreme end points; the distinction between late endosomes, multivesicular bodies, and fused endolysosomes is somewhat

fluid anyway). It is also advisable to check the movements on all available replicate maps (eg *Figure 4—figure supplement 2*).

Next, the organellar association as predicted by SVMs before and after treatment can be queried. To illustrate this, we have prepared an interactive database for the dynamic maps (control vs EGF treatment, *Supplementary file 6*). Proteins may undergo complete organellar translocations, as in the case of EGFR receptor; however, it is also possible that only a small proportion of the protein moves, and this may not be sufficient to change the overall SVM predictions. In other words, the MR plot is potentially more sensitive in detecting movement than a 'before-after' comparison of SVM predictions. Nevertheless, in cases of complete translocations, the SVM reveal the old and new compartments.

A further potentially revealing aspect is a protein's 'neighbourhood', ie proteins with similar fractionation profiles. In case of genuine translocations, the neighbourhood should change almost completely. The database shows the identity of the old and new neighbours, before and after EGF treatment, and this can be informative. For example, GRB2 as the query protein reveals SHC1 as a close neighbour after EGF treatment (but not before), consistent with their known complex formation triggered by EGFR activation.

## Identifying dynamic changes in nuclear, cytosolic and organellar pools

A central purpose of the organellar maps is to detect induced changes in protein subcellular localization. The global distribution profiles greatly add to this, as many translocations do not include two organellar compartments, but changes in organellar/cytosolic balance, or nuclear/cytosolic distribution. To determine any such global changes upon EGF treatment, we used the intensity data processed with the label free quantification (LFQ) algorithm (Group 3 data, see above). Data analysis was performed in Perseus (1.5.2.11; www.perseus-framework.org), separately for each compartment (organellar, nuclear cytosolic fractions). LFQ intensities were grouped (triplicate control vs triplicate EGF treatment experiments), log transformed, and filtered for a minimum of three valid values in at least one group. Missing values were then imputed from a normal distribution (downshift =2, width = 0.3). A (two-tailed) Student's t-test was then performed on the difference between control and EGF treatment for each protein. To identify stringent cut-offs for significant changes, we required a minimum 2-fold change in abundance, where the protein must constitute at least 10% of the total pool, either before or after EGF stimulation, in the compartment where it is shown to be changing (the total pool distribution was obtained from the intensity (Group 2) data, see above). An FDR controlled p-value cut-off was determined by treating the six 'Static' control maps as a mock comparative experiment, where three maps were assigned as mock-treated and three as control. The LFQ-analysis was then performed as for the control and EGF treatment maps. For each compartment, a p-value cut-off was determined such that no false positives would be detected in the mock experiment, but changes could still be detected in the EGF experiment (corresponding to an FDR of 0). This was possible for cytosol and organellar fractions; in the case of the nuclear fraction, with optimal cut-offs, 2 false positives are expected among the 13 detected significant changes in *Figure 4—figure supplement 2* (FDR=15%).

## Interpreting changes in global intensity distribution profiles

The LFQ-based statistical analysis of abundance changes in the nuclear, cytosolic and organellar fractions is very sensitive. However, it only reveals a fold change in fraction association, which then needs to be placed in the context of the protein's global distribution. (For example: The LFQ analysis shows that the cytosolic pool of GRB2 is decreased 2 fold upon EGF treatment (*Figure 4—figure supplement 2*); however, only the global profile reveals that this corresponds to a drop in the cytosolic pool from 90% to 40%, *Figure 4—figure supplement 3*). For the EGF experiment, all profile changes can be viewed in the interactive Dynamic database (*Supplementary file 6*). In most cases, the interpretation of changes in global profiles is straightforward (for example, a cytosolic depletion with concomitant increase in the organellar fraction, as for GRB2; or a nuclear exit with concomitant cytosolic increase, as for NR4A3). The exception is an intra-organellar translocation event involving organelles with very different levels of enrichment in the organellar fraction (see above, "*Interpretation of global intensity distribution profiles and their changes*"). A good example is the EGFR. Endosomal proteins have a much higher organellar enrichment in the organellar fraction than plasma

membrane proteins; a translocation from the plasma membrane to endosomes is thus accompanied by a highly characteristic shift from the nuclear to the organellar fraction (*Figure 4—figure supplement 3*). This case illustrates the importance of combining all levels of available data to interpret the observed changes (see also next paragraph).

## Combined interpretation of global and organellar changes

Combining the results from the organellar and global translocation detection may reveal cognate changes that strengthen the individual observations, and allow further interpretation. For example, the EGF-induced translocation of GRB2 to the endo-lysosomal system is accompanied by strong recruitment to the organellar fraction, and depletion from the cytosol; the number of GRB2 copies recruited to the organellar fraction is similar to the overall number of EGFR receptor proteins in the cell. In combination these observations are consistent with recruitment of GRB2 to EGFR, with approximately 1:1 stoichiometry, and movement to endosomes/lysosomes. In this way, a systematic, integrated analysis of changes can be performed (*Figure 5*).

## Rationale for uncoupling detection and interpretation of translocation events

The two-step procedure described above has several advantages over direct detection of translocation events, for example by comparing SVM organellar predictions before and after treatment. Firstly, our outlier test is much more sensitive and accurate, as it works on a lower data level, the quantitative mass spectrometry data (rather than on organellar predictions based on these data). SVM predictions, although very accurate and highly concordant between maps, necessarily introduce more noise into the data. This will lead to more false positives, especially for proteins with poor predictions in both sets, and proteins whose prediction may sway between two compartments. In contrast, with the outlier test presented here, it was possible to identify translocation events with an experimentally determined FDR of 0 for the EGF experiment. Furthermore, detectable changes in organellar predictions are restricted to complete translocation events (ie a protein fully translocating from one compartment to another). The outlier test, however, will also identify partial translocation events, ie small but consistent movements on the map, which do not lead to changes in the predicted compartment. Similarly, the SVM-based analysis is limited to proteins which have clear organellar predictions before and/or after treatment; proteins that are not assigned to an organelle cannot be identified as changing. In contrast, the outlier test will identify these changes, even if they cannot be fully interpreted afterwards.

## A guide to interpreting organellar maps

### Understanding the prediction output – reading organellar maps

The HeLa spatial proteome database (*Supplementary file 4* interactive; *Supplementary file 1* compact) and the interactive webpage (www.MapOfTheCell.org) provide the classification analysis in an easily accessible format. For the best interpretation of the predictions, we recommend that users consider all types of information available for a protein.

1. The highest-level prediction is the organellar classification. The prediction score provides guidance as to the confidence of the prediction; especially low scoring predictions warrant further investigation. Low scores can occur for at least three reasons: First, the protein may not be very abundant, resulting in fewer profiles, or lower quantification accuracy. Second, the protein may not have a clear single steady state organellar association, but multiple equally dominant associations, resulting in a mixed profile that does not closely match any compartment. Third, a protein may not at all be associated with any of the compartments mapped here.

2. The PCA-based maps (as shown on the webpage) provide complementary spatial information, and show where within a cluster the protein is located. For low-scoring proteins near the edge of a cluster, or proteins with an unexpected localization prediction, it may be useful to consider what other clusters are close by. In case the protein was mis-classified, the true association is most likely one of the neighbouring clusters. Similarly, some compartments are closely related, or may even be considered as 'snapshots' along a continuum (eg endosomes, late endosomes, and lysosomes; ER, ER-Golgi-intermediate compartment, and Golgi). Some proteins may be present in two 'adjacent' compartments, assume an intermediate position, and

therefore achieve only low prediction scores for either one. Again, the PCA maps may reveal if this is likely to be the case. As we provide six replicate maps, we recommend comparing a protein's position across all maps, to check for variability.

3. Furthermore, the neighbourhood analysis may provide very useful additional information. First, it shows which proteins (if any) have so closely related profiles that they may be forming a complex with the query; in this case, the organellar predictions for these proteins may also be of interest. Second, it provides an overview of the local map environment of the protein; if many of these proteins are established markers of the query protein's predicted class, it further supports its classification.

4. Finally, we advise users to consider the information on abundance and global protein distribution. Especially very abundant cytosolic proteins may be associated with various intracellular functions and locations (specific as well as non-specific), which may lead to broad (and difficult to interpret) profiles. Large nuclear and cytosolic pools suggest that the organellar fraction of these proteins is only minor, and the main function lies elsewhere in the cell.

## The Large Protein Complex (LPC) cluster

The organellar fraction, which is the basis for compartment assignment, contains numerous abundant non-membranous protein particles, such as ribosomes, proteasomes, the TRiC chaperone, the tripeptidyl peptidase (TPP2), and many more. These are sufficiently large and dense to pellet even at the relatively low speeds used to pellet membrane-bound organelles. This group is necessarily very heterogeneous, as the profiling behaviour of complexes is diverse, depending on size, density, association with other structures, etc. Nevertheless, as our PCA maps reveal, the group forms a cluster that is clearly distinguishable from membrane-bound compartments; we have named this the Large Protein Complex (LPC) cluster. Consistent with the notion that the LPCs contain no membranous compartments, there are almost no transmembrane domain proteins in this cluster (*Figure 2—figure supplement 1*). Within the cluster, individual protein complexes are clearly resolved in many cases (with some examples shown in *Figure 2—figure supplement 1*). For three reasons, we deliberately did not include any of these complexes in the SVM-based compartment predictions. First, many complexes have only few members, and provide insufficient cluster size for SVM classification. Second, the composition of many complexes is already established; any 'new' assignments are thus likely to have a very high FDR. Third, our HeLa spatial proteome database (*Supplementary file 4*) includes a 'neighbourhood analysis', which reveals which proteins are fractionating most closely with a query; this very convenient tool can (and has been) used to identify stable protein complexes, as well as their approximate stoichiometry (*Borner et al., 2014*). So although the LPC cluster is resolved into smaller clusters, we de-coupled this analysis from the SVM based compartment prediction.

Many of the complexes in the LPC cluster can transiently interact with membranes, including for example vesicle coat proteins such as clathrin, COPI, COPII. Organellar maps through fractionation profiling reveal only the majority steady state distribution of a protein; hence, if a complex is predominantly cytosolic, and only a relatively small proportion membrane-bound, the complex will be in the LPC class. Ribosomes are another case in point. Based on quantitative electron microscopy (*Attardi et al., 1969*), only 15–20% of ribosomes are attached to the ER membranes in HeLa cells, the remainder are 'free'. Consistent with these data, our analysis shows ribosomes in the LPC cluster, and not in the ER cluster. Furthermore, proteins may be affiliated with more than one complex, and show an 'in between' distribution that is not directly interpretable (for example SEC13, which has a known function in the nuclear pore complex, and in the COPII coat). We also noticed the presence of many nuclear proteins in the LPC cluster (eg histones). These proteins tend to have large nuclear pools in the membrane/nucleus/cytosol split, indicating that they are indeed predominantly nuclear in HeLa cells. Some of these proteins may well have a genuine small cytosolic pool; in addition, their presence may be due to a limited amount of nuclear breakage. We also noticed that many LPC proteins are RNA binding proteins, and it is very likely that they pellet as part of large assemblies on RNA. Finally, there are some proteins in the LPC class that are not known to be clearly associated with a particular complex, and the reasons for their presence in the LPC are currently unclear; a potential explanation is that some (abundant) soluble proteins may non-specifically stick to membranes or other complexes, giving rise to 'rogue' profiles. However, non-specific aggregation should not lead to consistent predictions of protein 'neighbourhoods', and we suggest that proteins with

close neighbours are more likely to be engaged in specific complexes. The bottom line is that any protein in the LPC fraction must be associated with a particle/structure large and dense enough to pellet at ~80,000 g (the maximum speed of the fractionation protocol).

## How to use the website www.mapofthecell.org

Please refer to *Supplementary file 10*, which shows a screenshot of the webpage, with annotations on how to use it.

## Acknowledgments

We would like to thank Matthias Mann for his generous support of this project. This work was funded by the German Research Foundation (DFG/Gottfried Wilhelm Leibniz Prize); the Louis-Jeantet Foundation; and the Max Planck Society for the Advancement of Science. We are very grateful to Korbinian Mayr, Igor Paron, and Gabriele Sowa for outstanding technical support. We would also like to thank Sebastian Schuck for a critical discussion of the manuscript, and all the members of the Mann Department for valuable feedback.

## Additional information

### Funding

| Funder | Grant reference number | Author |
|---|---|---|
| Deutsche Forschungsge-meinschaft | Gottfried Wilhelm Leibniz Preis | Georg HH Borner |
| Louis-Jeantet Foundation | | Daniel N Itzhak |
| Max-Planck-Gesellschaft | | Daniel N Itzhak Stefka Tyanova Jürgen Cox Georg HH Borner |

The funders had no role in study design, data collection and interpretation, or the decision to submit the work for publication.

### Author contributions

DNI, Conceptualization, Methodology, Experiments, Writing, Visualization, Investigation; ST, Software, Website; JC, Software; GHHB, Conceptualization, Methodology, Formal Analysis, Writing, Supervision

### Author ORCIDs

Georg HH Borner, http://orcid.org/0000-0002-3166-3435

## Additional files

### Supplementary files

• Supplementary file 1. The HeLa spatial proteome. A compact summary of organellar assignments and abundance information. Also includes the organellar markers used for classification.

• Supplementary file 2. Prediction performance. The performance and depth of all 12 maps reported in this study.

• Supplementary file 3. External validation of predictions. Contains the concordance analysis with external protein subcellular localization information. Our predictions are compared to UniProt annotations, and to a mouse cell line spatial proteome.

• Supplementary file 4. The complete HeLa protein subcellular localization database. An interactive database containing the full spatial information generated in this study, including localization, copy numbers, and neighbourhood analysis.

• Supplementary file 5. Anatomy of major organelles. The quantitative composition of three major compartments, ER, mitochondria, and plasma membrane.

• Supplementary file 6. EGF dynamics database. An interactive database showing organellar predictions and abundance changes induced by EGF.

• Supplementary file 7. EGF Translocation analysis. A summary of detected organellar and nuclear/cytosolic translocation events triggered by EGF.

• Supplementary file 8. Comparison of organellar profiling approaches. An overview of the features and requirements of Dynamic Organellar Maps and the LOPIT approach.

• Supplementary file 9. Raw data. The quantitative proteomics data used in this study (SILAC, intensity and LFQ data).

• Supplementary file 10. How to use www.MapOfTheCell.org.

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
