## [Decision Letter]

Thank you for submitting your article "Global, Quantitative and Dynamic Mapping of Protein Subcellular Localization" for consideration by *eLife*. Your article has been favorably evaluated by Randy Schekman as the Senior editor and three reviewers, one of whom is a member of our Board of Reviewing Editors.

The reviewers have opted to remain anonymous.

The reviewers have discussed the reviews with one another and the Reviewing Editor has drafted this decision to help you prepare a revised submission.

Summary:

This study describes a combined fractionation/SILAC approach for quantitative analysis of protein abundance and localization. The basic strategy is to use differential centrifugation of a 'light' labeled cell lysate and compare it to a 'heavy' reference. The quantitative profiles of abundances across the different fractions provide a signature for that protein's cellular location. The procedure can be tracked in differentially treated cells to monitor changes in location, with the quantitative nature of the entire process revealing estimates of absolute protein abundances. The authors have also developed a set of interactive tools, both a web-based graphical user interface, and interactive excel-based worksheets that enable rapid examination of proteins of interest and provide a wealth of data including subcellular localization as well as estimations of absolute copy numbers. The authors provide in-depth discussions of each component of the actual mass-spec method as well as all subsequent bioinformatic analyses within the supplemental materials, making duplication of the approach, at least in principle, achievable by any lab with unfettered access to high resolution mass spectrometry instrumentation.

Essential revisions:

All of the referees found the manuscript well written and enjoyable to read, and the tools described therein a valuable resource for the research community. In the initial referee comments, two of us felt that some additional validation to highlight the method's capacity to discover new biology would be useful. During the ensuing discussion, it was agreed that such additional experimentation was probably beyond the scope of a "Tools" submission, but that if such data were available, it would be a welcome addition to the paper. One example of such validation might be to verify a couple of the previously unknown changes in localization induced by EGF stimulation that emerged from the existing datasets.

The main essential revisions center around requests for greater explanation of the method and clarification of the precise methods of analysis. These should be possible by modifications to the text. Excerpts of the relevant requests to be addressed are as follows:

1) The chosen set of validated markers is extremely important to evaluate the performance of this approach. The authors acknowledge as much in their nice description of how markers were chosen. The authors claim a 94.7% predication accuracy based on these makers. However, an unknown subset of these markers were chosen based on their ability to localize to discrete subcellular structures based on pilot experiments using the same proteomic approach. This generates a circular logic where markers that already show specific locations validate the same markers. I understand the need to choose appropriate markers, and the currently available list of markers may or may not be sufficient for such approaches, but the authors should be more explicit about how this was done in the main text. To their credit, the authors are forthright about this in the supplemental information.

2) One of the referees noted that many of their favorite proteins were either not present or not usable in the Excel file, apparently due to their low abundance. In addition, some nuclear proteins queried were designated as large protein complexes – which they may be – or maybe they are just aggregating in the experiment. Either way, a discussion about the problems associated with detection of low abundance proteins would be appreciated here, as parts of the discussion are suggesting that near 'comprehensive coverage' is achieved – a claim of which the referee was uncertain.

3) The dynamic organellar mapping was of particular interest, but it was hard to follow exactly how this was done (further explanation or a flow chart would be helpful) and how many mass spec runs this took – if it was done in triplicate at two time points, this seems like a lot of ms time (6 independent runs – is that about 2 – 3 weeks on a Q-exactive?) – this should be included in the Results section – could this be improved with TMT?

---

## [Author Response]

Essential revisions:

All of the referees found the manuscript well written and enjoyable to read, and the tools described therein a valuable resource for the research community. In the initial referee comments, two of us felt that some additional validation to highlight the method's capacity to discover new biology would be useful. During the ensuing discussion, it was agreed that such additional experimentation was probably beyond the scope of a "Tools" submission, but that if such data were available, it would be a welcome addition to the paper. One example of such validation might be to verify a couple of the previously unknown changes in localization induced by EGF stimulation that emerged from the existing datasets.

We concur with referees’ decision that further experimental validation of the results would be beyond the scope of this study. We will therefore not include any further experimental results.

*The main essential revisions center around requests for greater explanation of the method and clarification of the precise methods of analysis. These should be possible by modifications to the text. Excerpts of the relevant requests to be addressed are as follows:*

1) The chosen set of validated markers is extremely important to evaluate the performance of this approach. The authors acknowledge as much in their nice description of how markers were chosen. The authors claim a 94.7% predication accuracy based on these makers. However, an unknown subset of these markers were chosen based on their ability to localize to discrete subcellular structures based on pilot experiments using the same proteomic approach. This generates a circular logic where markers that already show specific locations validate the same markers. I understand the need to choose appropriate markers, and the currently available list of markers may or may not be sufficient for such approaches, but the authors should be more explicit about how this was done in the main text. To their credit, the authors are forthright about this in the supplemental information.

We agree with the referee that the choice of organellar markers is critical for the success of the organellar assignment by supervised learning. We also note that the referee is essentially happy with our detailed explanation of how and why markers were chosen, as presented in the Methods (subsection “Organellar marker set generation”). The referee’s main concerns thus pertain to estimating the overall prediction accuracy based on marker protein predictions, and the lack of explaining the choice of markers in the main text.

We agree that the 94.7% percent correctly predicted markers is not a true estimate of prediction accuracy for non-marker proteins. Nevertheless, prediction accuracy of the markers can serve as a useful proxy, since it is evaluated with “leave-one-out” cross-validation: for a set of 1,000 marker proteins, models are built and predictions made 1,000 times, using a different combination of 999 marker proteins each time. This essentially simulates the prediction of proteins that are not in the training set, and ensures that the support-vector machine models have maximum predictive value for non-maker proteins. The marker protein prediction accuracy is hence an indicator of the overall performance for those new predictions that fall within the boundaries defined by the markers.

It would be possible to choose completely non-overlapping sets of makers (e.g. only those close to the cluster center), but this would be of little value for organellar predictions, since much unchartered space would remain in between clusters. We therefore attempted to define the outermost boundaries of each organellar cluster, by finding literature-accepted maker proteins at the interfaces between two clusters, as judged by visual inspection of PCA plots from pilot maps. In many cases, we even chose markers that visibly ‘interdigitate’ the clusters, to make the boundaries as useful as possible. The aim was not to get the best scores for marker prediction accuracy, but to obtain the best boundaries, i.e. the ones with the best predictive value for further proteins.

In the absence of a suitable independent marker set (which does not exist), true prediction accuracy is difficult to gauge. We therefore validated our non-marker predictions through multiple comparisons with external data (e.g. uniprot annotations, in silico prediction of targeting sequences, Christoforou et al., 2016), and observed excellent agreement in all cases. For example, the prediction concordance with the Christoforou study is 91.6%. Furthermore, there is an independent database for experimentally validated mitochondrial proteins (“MitoCarta 2.0” (Calvo et al., 2016)), which we have now cross-referenced against our predictions while this manuscript was under review. Of the 658 proteins we predict as mitochondrial, 638 are validated mitochondrial proteins; this corresponds to 97% agreement, close to the 98.8% accuracy suggested by the marker protein prediction (Table 1). We have added the concordance with MitoCarta to the Results section, as further external validation of our predictions (subsection “A database of protein subcellular localization”, first paragraph).

In conclusion, we agree with the referee that marker protein predictions do not reflect true prediction accuracy; but since our markers were chosen to model cluster boundaries as well as possible, we are convinced that it is still worth reporting these figures. They also give a measure of differences in prediction accuracy for different organelles (Table 1), and thus indicate which organelles are better resolved than others. In the manuscript text, we carefully distinguish between “marker prediction accuracy” and “true prediction accuracy”. Furthermore, we have added the following sentence to the Results section, to clarify the difference explicitly:

“The mean prediction accuracy for marker proteins was 94.7% (with leave-one-out cross validation), demonstrating the high level of organellar resolution achieved ([Supplementary-material SD2-data]). While marker prediction accuracy does not provide a direct measure of overall prediction accuracy, it nevertheless serves as a useful estimate (see Methods for further details).”

We have also added a detailed discussion of prediction accuracy to the Methods (subsection “Estimating overall prediction accuracy”).

In the Abstract, we have replaced the estimated overall prediction accuracy (previously quoted at 96% for all marker proteins) with the average for membrane-bound organelles (92%), as the more relevant (and more conservative) estimate.

Following the referee’s other suggestion, we have added a description of how the marker proteins were chosen to the Results section:

“For the rigorous assignment of proteins to organelles, we used a support vector machine (SVM)-based supervised learning approach. […] Where necessary, we specifically chose further established markers near the edges of organellar clusters, as these are particularly important for defining boundaries. We applied SVM classification to all six maps individually.”

We have also augmented the Materials and methods section on marker set generation with some of the arguments presented above (subsection “Organellar marker set generation”).

2) One of the referees noted that many of their favorite proteins were either not present or not usable in the Excel file, apparently due to their low abundance. In addition, some nuclear proteins queried were designated as large protein complexes – which they may be – or maybe they are just aggregating in the experiment. Either way, a discussion about the problems associated with detection of low abundance proteins would be appreciated here, as parts of the discussion are suggesting that near 'comprehensive coverage' is achieved – a claim of which the referee was uncertain.

We accept the referee’s point that the database does not provide complete coverage of the human proteome. First, only one cell type (HeLa) was investigated; many tissue- or cell-type specific proteins are not expressed in HeLa cells. Our database of (8,700 entries) covers the majority of the HeLa proteome (estimated at around 10,000). This includes over 2,000 proteins that are almost exclusively cytosolic and/or nuclear; consequently, these have no organellar profiles, and their localization is directly inferred from their global (cytosolic/nuclear) distribution. Since we applied very stringent quality criteria to the mass spectrometry data for the organellar profiling analysis, there are also several hundred ‘orphaned’ membrane-associated proteins that were not sufficiently abundant for accurate mapping. Our two-tiered approach nevertheless allowed us to provide some spatial information for these proteins, including the nuclear/organellar/cytosolic distribution as well as copy number and cellular concentration. At the individual protein localization level, our coverage is hence substantial, albeit not complete. On the other hand, our database accounts for the vast majority of total cell protein mass; the cumulative protein mass plots in Figure 3 all plateau, suggesting that further identified proteins would be of low abundance, and add minimally to overall organellar mass and copy number composition. In that sense, our coverage is ‘comprehensive’, which is critically important for the organellar modelling shown in Figure 3.

In the main text, we have corrected the use of the phrase ‘comprehensive coverage’ where necessary. In the context of organellar modelling, we have kept it, since in our opinion it is justified. In other cases, we have toned down our statements, as suggested by the referee.

Furthermore, we appreciate the Referee’s concern that future users of our database may be left confused as to why some proteins are not as extensively annotated as others. To ameliorate this problem, we have added a new ‘FAQ’ section to the interactive Excel database ([Supplementary-material SD4-data]), to explain why different levels of information are available for some proteins, and why some proteins may not be present in the database:

“I have entered the correct gene name, but my query is still not found – why not?

[…] If your query falls into this category, it is either not very well expressed in HeLa cells, or its organellar pool is very small.”

The referee raises another important point, namely why many nuclear proteins are assigned to the ‘large protein complex’ (LPC) cluster. Firstly, although they have a prediction for LPC, the large nuclear pool of these proteins, as determined in the global distribution, indicates that they are predominantly nuclear. Their presence in the LPC cluster suggests that some may well have a genuine cytosolic pool; alternatively, their presence may also be due to a small amount of nuclear breakage. Either way, these proteins behave like large protein complexes during fractionation, defined as pelleting substantially at 80,000 g. While we cannot rule out that in some cases this may be due to aggregation, non-specific aggregation would not lead to consistent predictions of protein ‘neighbourhoods’, and we suggest that proteins with close neighbours are indeed likely to be engaged in specific complexes. Furthermore, we noticed that many LPC proteins are RNA binding proteins, and it is very likely that they pellet as part of large assemblies on RNA. In the original submission, we devoted a paragraph to the interpretation of the LPC cluster (Methods). We have now augmented this paragraph to include a new section dedicated to the presence of nuclear proteins in the LPC cluster.

*3) The dynamic organellar mapping was of particular interest, but it was hard to follow exactly how this was done (further explanation or a flow chart would be helpful) and how many mass spec runs this took – if it was done in triplicate at two time points, this seems like a lot of ms time (6 independent runs – is that about 2 – 3 weeks on a Q-exactive?) – this should be included in the Results section – could this be improved with TMT?*

We agree with the referee that the details of the experimental setup may be hard to follow. To clarify: The dynamic maps were applied to control cells, and cells treated with EGF for 20 min, each in triplicate (six maps total). As detailed in the Methods, we suggest two alternative mass spec measuring strategies to analyse maps, which afford different depths. The ‘deep’ protocol requires 60h of measuring time per map, and was used for the ‘static’ HeLa maps to construct the database. The ‘fast’ protocol requires only 20h of measuring time per map. We used this ‘fast’ protocol for the dynamic EGF maps (Figure 1–Figure 5). Hence, the entire dynamic organellar maps experiment required 6x20h = 120h (5 days) of mass spec measuring time, a comparatively moderate amount.

To help readers understand the experimental design of the dynamic organellar maps, we have added a new supplemental figure detailing the complete design, colour-coded according to the different branches of analysis (Figure 1—figure supplement 1). In addition, our Materials and methods section includes a small table (subsection “Mass spectrometry measuring time requirements and map dept”) listing the mass spec measuring time requirements, and we have added the following sentence to the Discussion:

“The entire experiment (triplicate comparisons, six maps) required only five days of mass spectrometry measuring time.”

The referee’s suggestion to improve throughput by switching to TMT labelling is certainly interesting. TMTs offer sample multiplexing, and thus the potential to run a complete map as one sample; on the other hand, the notorious TMT ‘ratio compression’ issue may seriously compromise the accuracy of maps. While MS3 approaches largely avoid ratio compression, they require special mass spectrometry instrumentation (Fusion/Lumos), which is not as widely available as the Q Exactive type instrument used by ourselves. In addition, the MS3 approach compromises sequencing depth, and thus requires more extensive fractionation; this somewhat offsets the advantage of multiplexing. Hence, future experiments will be required to test the suitability of TMTs for our approach. We have added a sentence to the Discussion, to point out this potentially promising future avenue:

[…] “In addition, mass tagging is in principle compatible with our approach, too, and may thus extend its range of applications in future.”